# The Utrecht Finite Volume Ice-Sheet Model: UFEMISM (version 1.0)

Constantijn J. Berends[1], Heiko Goelzer[1,2,3], Roderik S. W. van de Wal[1,4]

[1] Institute for Marine and Atmospheric research Utrecht, Utrecht University, Utrecht, 3584 CC, The Netherlands
[2] Laboratoire de Glaciologie, Université Libre de Bruxelles, Brussels, Belgium
[3] NORCE Norwegian Research Centre, Bjerknes Centre for Climate Research, Bergen, Norway
[4] Faculty of Geosciences, Department of Physical Geography, Utrecht University, Utrecht, The Netherlands

*Correspondence to*: Constantijn J. Berends (c.j.berends@uu.nl)

**Abstract.** Improving our confidence in future projections of sea-level rise requires models that can simulate ice-sheet evolution both in the future and in the geological past. A physically accurate treatment of large changes in ice-sheet geometry requires a
proper treatment of processes near the margin, like grounding line dynamics, which in turn requires a high spatial resolution in that specific region, so that small-scale topographical features are resolved. This leads to a demand for computationally efficient models, where such a high resolution can be feasibly applied in simulations of $10^5 - 10^7$ yr in duration. Here, we present and evaluate a new ice-sheet model that solves the SIA and SSA approximations of the stress balance, including a heuristic rule for the grounding-line flux, on a dynamic adaptive mesh. Mesh resolution can be configured to fine only at
specified areas, such as the calving front or the grounding line, as well as specified point locations such as ice-core drill sites. This strongly reduces the number of grid points where the equations need to be solved, increasing the computational efficiency. We show that the model reproduces the analytical solutions or model intercomparison benchmarks for a number of schematic ice-sheet configurations, indicating that the numerical approach is valid. Because of the unstructured triangular mesh, the number of vertices increases less rapidly with resolution than in a square-grid model, greatly reducing the required computation
time for high resolutions. A simulation of all four continental ice sheets during an entire 120 kyr glacial cycle, with a 4 km resolution near the grounding line, is expected to take 100 – 200 wall clock hours on a 16-core system (1,600 – 3,200 core hours), implying that this model can be feasibly used for high-resolution paleo-ice-sheet simulations.

## 1 Introduction

The response of the Greenland and Antarctic ice sheets to the warming climate forms the largest uncertainty in long-term sea-
level projections (e.g. Oppenheimer et al., 2019; van de Wal et al., 2019). Since the dynamical evolution of ice sheets has components that are slow compared to the human time scale, observational evidence alone cannot sufficiently reduce this uncertainty. Instead, reconstructions of the evolution of ice sheets during the geological past are required to improve our understanding of the long-term evolution of these systems and the constraints this provides for future ice-sheet retreat. Recent work has focused on using ice-sheet models and climate models, with varying degrees of inter-model coupling, to reproduce
different periods of the geological past, with climates that have been both significantly warmer and colder than the present

(e.g. Abe-Ouchi et al., 2013; Pollard et al., 2013; DeConto and Pollard, 2016; Stap et al., 2017; Berends et al., 2018, 2019; Willeit et al., 2019). These studies have highlighted and reemphasized the importance of understanding, and properly modelling, the different physical interactions between ice sheets, sea level, the solid Earth, and the regional and global climate. Since several of these processes become relevant only when significant changes in ice-sheet geometry occur, studying them requires very long ($10^5 - 10^7$ yr) ice-sheet model simulations.

The dynamics of the Antarctic ice sheet at present are strongly influenced by the presence of floating ice shelves (Pattyn, 2018). Different studies have investigated the physical processes affecting these ice shelves, including surface melt induced by atmospheric processes (Bevan et al., 2017; Kuipers Munneke et al., 2018), bottom melt induced by intrusion of warm ocean water (Depoorter et al., 2013; Lazeroms et al., 2018, Reese et al., 2018), brittle fracturing of ice cliffs (Pollard et al., 2015), and the response of the grounding line to changes in sea level and bedrock elevation (Gomez et al., 2013; Barletta et al., 2018). Ice dynamics around the grounding line have been of particular interest, with some studies suggesting that a very high model resolution (< 100 m) is needed to accurately resolve the physical processes involved (Schoof, 2007; Gladstone et al., 2012; Pattyn et al., 2012, 2013). Since this not achievable for palaeo-applications, different approaches have been proposed where semi-analytical solutions (Schoof, 2007; Tsai et al., 2015) are implemented as boundary conditions in numerical models, to maintain physical accuracy at lower resolutions (Pollard & DeConto, 2012; Pattyn, 2017) of 1 – 40 km. However, while these approaches achieve good results in situations without buttressing (Pattyn et al., 2013), their performance in simulating realistic ice shelves, where buttressing is usually a significant factor, is poor (Reese et al., 2018). Furthermore, a coarse model resolution can affect simulated ice-sheet evolution not just through numerical errors, but also by insufficiently resolving small-scale topographical features such as fjords and mountains (Cuzzone et al., 2019).

Ice-sheet model computation time increases rapidly with model resolution, due to the increasing number of grid points, the decreasing time step required for numerical stability, and the increasing number of topographic features that are resolved. This means that the need for both a high model resolution and long simulations results in a computational paradox. Most research groups working on paleo-ice-sheet simulations consider the uncertainties in paleoclimate reconstructions used as forcing to be much larger than the physical inaccuracy resulting from a low model resolution. The ice-sheet models used by these groups therefore typically have a low-resolution, fixed grid, and solve a simplified version of the Navier-Stokes equations, which makes them very computationally efficient, and relatively easy to compile, run, and modify (e.g. SICOPOLIS; Greve et al., 2011; PISM; Winkelmann et al., 2011; ANICE; de Boer et al., 2014; f.ETISh; Pattyn, 2017; GRISLI; Quiquet et al., 2018; CISM; Lipscomb et al., 2019; Yelmo; Robinson et al., 2020). Paleo-simulations of the Antarctic ice sheet with such models have used resolutions of e.g. 10 km (DeConto and Pollard, 2016), 32 km (Robinson et al., 2020), 40 km (Berends et al., 2019), 80 km (Willeit et al., 2019) or 110 km (Abe-Ouchi et al., 2013), too low to properly capture grounding line dynamics. A few existing models, which are mainly intended for relatively short ($10^1$-$10^3$ yr) simulations, use high-resolution, static adaptive grids ("static" meaning that the grid is adapted to the initial ice-sheet geometry, but is not updated during a simulation, e.g.

ISSM; Larour et al., 2012; Elmer/Ice; Gagliardini et al., 2013; MALI; Hoffman et al., 2018) or even dynamic adaptive grids ("dynamic" meaning that the grid is adapted to the evolving ice-sheet geometry during a simulation, e.g. BISICLES (Cornford et al., 2013). These more sophisticated models solve for more terms in the Navier-Stokes equations (either the higher-order Blatter-Pattyn [Pattyn, 2003] approximation, or even the full Stokes system) using finite element methods, making them more

physically accurate. While recent developments have improved their computational efficiency enough to make small-scale (single ice-sheet basin, $10^4$ yr) paleo-simulations feasible (Cuzzone et al., 2018, 2019), they tend to be too computationally demanding for the $10^5$-$10^7$ yr simulations needed for paleo-ice-sheet simulations.

Here, we present and evaluate a new ice-sheet model that constitutes a compromise between these two families: the Utrecht

Finite Volume Ice-Sheet Model (UFEMISM). It combines the SIA and SSA simplifications of the stress balance used in most paleo-ice-sheet models with a dynamic adaptive grid, which allows it to achieve a high (< 5 km) resolution near the grounding line, while retaining the computational speed required for feasibly performing long paleo-ice-sheet simulations. This makes it especially useful for studying the impact of ice-sheet – solid Earth – sea-level interactions on long-term ice-sheet evolution. In Sect. 2, we provide a brief description of the physical equations for ice dynamics and thermodynamics that are solved by

the model, as well as a description of the dynamic adaptive grid upon which those equations are solved. In Sect. 3, we present results from a number of benchmark experiments performed with UFEMISM, showing that the model output agrees well with different analytical solutions, as well as with output from other ice-sheet models. In Sect. 4, we present results from a detailed analysis of the computational performance of the model.

## 2 Model description

### 2.1 Overview

UFEMISM is a variable-resolution ice-sheet model. Flow velocities for grounded ice are calculated using the shallow ice approximation (SIA; Morland and Johnson, 1980), while the shallow shelf approximation (SSA; Morland, 1987) is used both for calculating flow velocities for floating ice, and as a "sliding law" for grounded ice, using the hybrid approach by Bueler and Brown (2009). These equations are discretised and solved on a dynamic adaptive grid (also called a mesh), which is

generated and updated internally based on the modelled ice-sheet geometry. The resolution can vary from as coarse as 200 km over open ocean or 100 km over the interior of a large ice-sheet, to as fine as 1 km at the grounding line, with the precise numbers specified at run-time through a configuration file. Using a finite volume approach (hence the name), ice velocities and fluxes are calculated on cell boundaries, similar to the "staggered" Arakawa C grid used in many ice models based on finite differences (Arakawa and Lamb, 1977). By explicitly calculating the mass of ice moved between individual vertices in

every time step, this approach guarantees conservation of mass. The model is thermomechanically coupled, solving for the diffusion and advection of heat, which enters the ice sheet through the surface and base, as well as through strain heating. For this purpose, the ice sheet is divided vertically into 15 unequally spaced layers (also configurable), decreasing in thickness

near the base, where most of the deformation takes place. The resulting three-dimensional englacial temperature field is used to determine the ice viscosity.

## 2.2 Unstructured triangular mesh

The main distinguishing feature of UFEMISM with respect to other paleo-ice-sheet models is the unstructured triangular mesh
on which the physical equations are solved. UFEMISM includes its own mesh generation code, which is based on an extended version of Ruppert's algorithm (Ruppert, 1995). This algorithm is discussed in more detail in Appendix E. Shown in Fig. 1 are four meshes generated for both Antarctica and Greenland, based on present-day ice-sheet geometry (BedMachine Greenland version 3, Morlighem et al., 2017, and BedMachine Antarctica version 1, Morlighem et al., 2019), with maximum ice-margin (including both the grounding line and the calving front) resolutions of 100, 30, 10 and 3 km, respectively. For Antarctica, two
high-resolution locations are included: one at the South Pole, and one at the EPICA Dome C drill site.

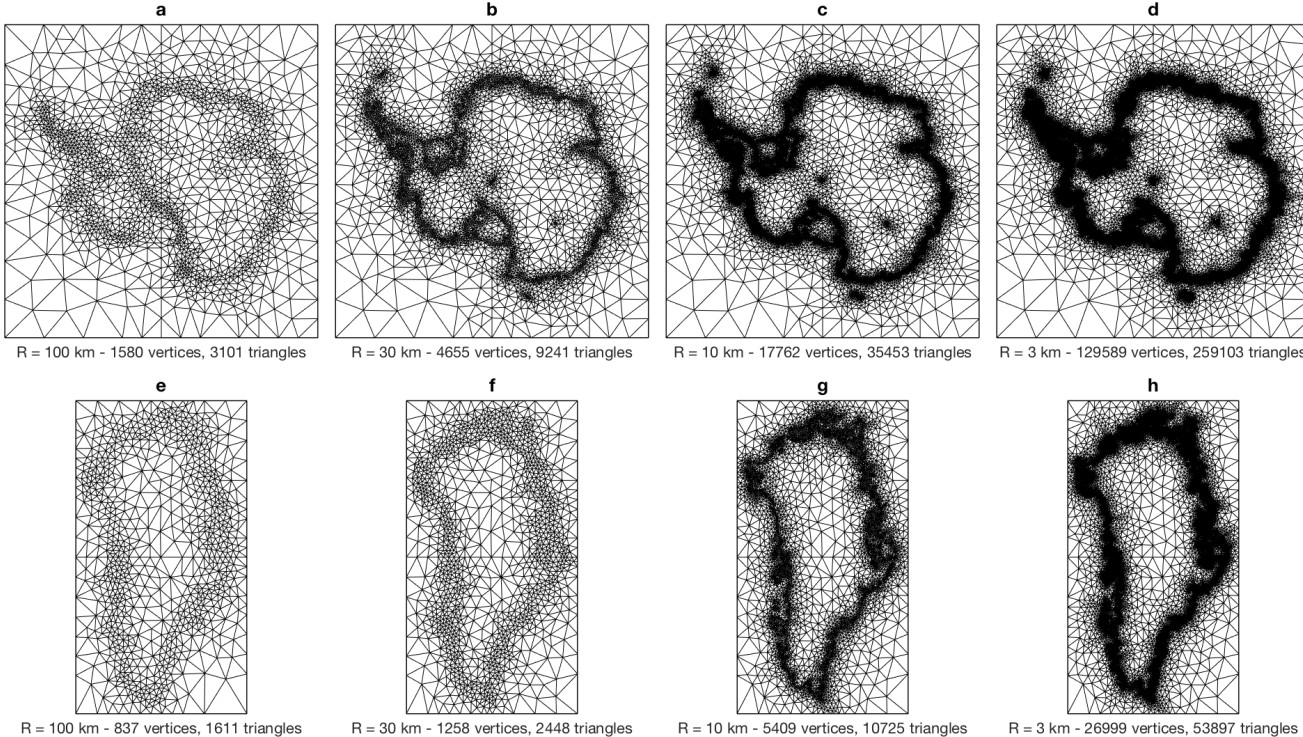

**Figure 1: Meshes generated for Antarctica (a-d) and Greenland (e-h), based on present-day ice-sheet geometry, with maximum ice-margin (including grounding line and calving front) resolutions of 100, 30, 10 and 3 km, respectively. For Antarctica, two high-resolution locations are included: one at the South Pole, and one at the EPICA Dome C drill site. Both have been prescribed the**
**same resolution as the ice margin.**

Since the ice margin, the calving front and the grounding line are one-dimensional regions, increasing the desired resolution only over these regions, results in a total number of vertices and triangles that scales almost linearly with this resolution (though not quite, as increasing the resolution resolves more geographical features, increasing the length of the lines). How the number

of vertices and the computational speed of the model scale with the prescribed resolution, is investigated in more detail in Sect. 4.

## 2.3 Ice dynamics

UFEMISM uses the hybrid SIA/SSA approximation to the stress balance developed by Bueler and Brown (2009). In this
approximation, ice velocities over land are calculated using the SIA, whereas ice velocities for floating ice, as well as sliding velocities on land, are calculated using the SSA. The two velocity fields are then added together, following the approach developed for PISM by Winkelmann et al. (2011), who argued that the weighted average used by Bueler and Brown (2009) introduces a new degree of freedom, while not appreciably changing the solution.

The SIA relates the (depth-dependent) ice velocities $u(z), v(z)$ directly to local ice thickness $H$, the temperature-dependent viscosity $A(T^*)$, and the surface slopes $\frac{\partial h}{\partial x}, \frac{\partial h}{\partial y}$ (with the dependence on $x,y,t$ in all variables left out for ease of notation):

$$D(z) = 2(\rho g H)^n |\nabla h|^{n-1} H \int_0^z A(T^*)\zeta^n d\zeta,$$ (1)

$$u(z) = D(z)\frac{\partial h}{\partial x},$$
$$v(z) = D(z)\frac{\partial h}{\partial y}.$$ (2)

The depth-dependent ice diffusivity $D(z)$ is defined as a function of the ice density $\rho$, gravitational acceleration $g$, ice thickness $H$, surface gradient $\nabla h$, Glen's flow law exponent $n$, and the temperature-dependent ice flow factor $A(T^*)$. For a comprehensive derivation of Eqs. 1 and 2, see e.g. Bueler and Brown (2009).

A concrete version of the SSA stress balance, expressed in terms of the vertically averaged horizontal ice velocities $u, v$, is given by Bueler and Brown (2009). Here, subscripts denote partial derivatives, e.g. $u_x = \frac{\partial u}{\partial x}$:

$$\frac{\partial}{\partial x}\left[2\bar{v}H(2u_x + v_y)\right] + \frac{\partial}{\partial y}\left[\bar{v}H(u_y + v_x)\right] - \frac{\tau_c u}{|\boldsymbol{u}|} = \rho g H h_x,$$ (3a)

$$\frac{\partial}{\partial x}\left[\bar{v}H(u_y + v_x)\right] + \frac{\partial}{\partial y}\left[2\bar{v}H(u_x + 2v_y)\right] - \frac{\tau_c v}{|\boldsymbol{u}|} = \rho g H h_y.$$ (3b)

The first two terms on the left describe the extensional stresses, also called "membrane stresses". The third term describes the basal shear stress for a Coulomb-type friction law, which is commonly used in hybrid SIA/SSA models since the vanishing
friction at the grounding line yields better results than the discontinuous Weertman-type friction law. The right-hand side describes the gravitational driving stress. The vertically averaged ice viscosity $\bar{v}$ is described by MacAyeal (1989) as a function of ice velocity:

$$\bar{v} = \frac{1}{2} \int_b^h A(T^*)^{\frac{-1}{n}} dz \left[ u_x^2 + v_y^2 + u_x v_y + \frac{1}{4} \left( u_y + v_x \right)^2 \right]^{\frac{1-n}{2n}}. \tag{4}$$

In order to ensure proper grounding line migration, a semi-analytical solution for grounding line flux with a Coulomb-type sliding law (Tsai et al., 2015) is applied as a boundary condition for the SSA:

$$q_g = Q_0 \frac{8A(\rho g)^n}{4^n \tan \varphi} \left( 1 - \frac{\rho_i}{\rho_w} \right)^{n-1} H_g^{n+2} \tag{5}$$

The way this solution is implemented is described in detail in Appendix C.

In order to solve these equations numerically, these different quantities need to be discretised on a grid. In many ice models, this is done using a combination of a standard central differencing scheme, and a "staggered" grid, where either the ice thickness $H$ is averaged and the surface slopes $\frac{\partial h}{\partial x}$ and $\frac{\partial h}{\partial y}$ are differenced (Type I models; Huybrechts et al., 1996), or the other way around (Type II models). UFEMISM is, in this sense, a Type I model. The discretisation of spatial derivatives on an unstructured triangular mesh is derived in Appendix A. Ice thickness changes over time are calculated using a finite volume

approach: by calculating both the diffusivity $D$ and surface slopes $\frac{\partial h}{\partial x}$ and $\frac{\partial h}{\partial y}$, and the resulting ice velocities $u$ and $v$, on the boundaries between vertices (using the "staggered mesh" approach described in Appendix A), ice fluxes between individual vertices are calculated explicitly, and moved from one vertex to the other in every time step. This guarantees conservation of mass. The finite volume approach is explained in more detail in Appendix B.

### 2.4 Thermodynamics

UFEMISM uses a mixed implicit-explicit finite differencing scheme with a fixed time step to solve the heat equation in a flowing medium:

$$\frac{\partial T}{\partial t} = \frac{k}{\rho c_p} \nabla^2 T - \boldsymbol{u} \cdot \boldsymbol{\nabla} T + \frac{\Phi}{\rho c_p}, \tag{6}$$

$$\Phi = 2 \left( \dot{\varepsilon}_{xz} \tau_{xz} + \dot{\varepsilon}_{yz} \tau_{yz} \right) = -\rho g (h - z) \left[ \frac{\partial u}{\partial z} \frac{\partial h}{\partial x} + \frac{\partial v}{\partial z} \frac{\partial h}{\partial y} \right] \tag{7}$$

The three terms on the right-hand side of Eq. 6 respectively represent diffusion, advection and strain heating (for grounded ice only; strain heating for shelves is at present not included). In UFEMISM, horizontal diffusion of heat is neglected, simplifying Eq. 6 to:

$$\frac{\partial T}{\partial t} = \frac{k}{\rho c_p} \frac{\partial^2 T}{\partial z^2} - \boldsymbol{u} \cdot \boldsymbol{\nabla} T + \frac{\Phi}{\rho c_p}, \tag{8}$$

This equation is discretised on an irregular grid in the vertical direction; all vertical derivatives are discretised implicitly, whereas horizontal derivatives are discretised explicitly. This mixed approach has a long history of use in palaeo-ice-sheet models (e.g. Huybrechts, 1992; Greve, 1997), as it is numerically stable (since both the steepest gradients and shortest grid

distances are in the vertical direction), relatively easy to implement, and fast to compute. Using 15 layers in the vertical direction, a time step of 10 years is typically sufficient to maintain numerical stability for the range of resolutions investigated here. The iterative scheme for solving this equation is derived in Appendix C.

## 3 Model verification and benchmark experiments

In order to verify the numerical solution to the ice dynamical equations, we performed several benchmark experiments, where we compare our model output to analytical solutions (Halfar, 1981; Bueler et al., 2005), and to results from the EISMINT intercomparison exercise (Huybrechts et al., 1996), and finally for the SSA part of the solution to the MISMIP experiments (Pattyn et al. 2012).

### 3.1 Verification using analytical solutions

For several schematic, simplified ice-sheet configurations, analytical solutions exist for the time evolution of the ice sheet. One of these was derived by Halfar (1981), describing a "similarity solution" for the time evolution of a radially symmetrical, isothermal ice sheet lying on top of a flat bed, with a uniform zero mass balance. For an ice sheet which, at time $t_0$, has a thickness at the dome $H_0$ and margin radius $R_0$, the time-dependent solution to the SIA, with Glen's flow law exponent $n$, reads:

$$H(r,t) = H_0 \left(\frac{t_0}{t}\right)^{\frac{2}{5n+3}} \left[1 - \left(\left(\frac{t_0}{t}\right)^{\frac{1}{5n+3}} \frac{r}{R}\right)^{\frac{n+1}{n}}\right]^{\frac{n}{2n+1}}, \tag{9}$$

$$t_0 = \frac{1}{(5n+3)\Gamma}\left(\frac{2n+1}{n+1}\right)^n \frac{R_0^{n+1}}{H_0^{2n+1}}, \Gamma = \frac{2A}{5}(\rho g)^n \tag{10}$$

Since the surface mass balance is zero, any change in ice thickness is caused only by ice dynamics, making this a useful experiment for verifying ice-sheet model numerics.

We performed several simulations with UFEMISM of an ice-sheet that starts at $t = t_0$ with the Halfar solution for $H_0 = 5000$ m, $R_0 = 300$ km, $A = 10^{-16}$ Pa³ yr⁻¹, and $n = 3$. The ice-margin resolutions for the different simulations were set to 50, 16, 8,
and 4 km. Shown in Fig. 3 are transects of the simulated ice sheet at different points in time, compared to the analytical solution.

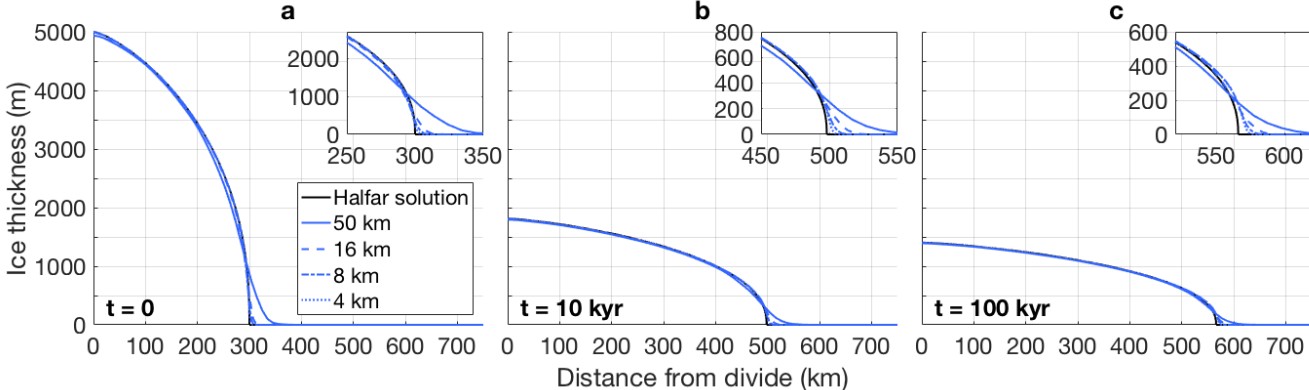

Figure 3: The evolution through time of a schematic, radially symmetric, isothermal ice sheet, as simulated by UFEMISM at different ice-margin resolutions, compared to the Halfar solution starting at $t = t_0$. The small sub-panels in the top-right corner of the panels show a zoomed-in view of the ice margin, showing how the simulated ice margin converges to the analytical solution as the model resolution increases.

UFEMISM reproduces the analytical solution well, with the largest errors occurring at the margin, and decreasing with resolution. As was shown by Bueler et al. (2005), this is the case for all spatially discrete SIA models, and is due to the fact that such models are intrinsically unable to reproduce the infinite surface slope at the margin predicted by the continuum model. They also show that these errors do not "corrupt" the model solution over the interior. This matches our results, where the modelled ice-sheet interior after 100,000 years is still close to the analytical solution. At that time, the modelled ice-sheet margin at 4 km resolution differs from the analytical solution by about 10 km.

A generalisation of the solution by Halfar (1981), applicable to problems including a simple elevation-dependent accumulation rate, was derived by Bueler et al. (2005). For an accumulation given by:

$$M_\lambda(r,t) = \frac{\lambda}{t} H(r,t),$$

(11)

the solution for the ice thickness over time reads:

$$H_\lambda(r,t) = H_0 \left(\frac{t_0}{t}\right)^\alpha \left\{ 1 - \left[ \left(\frac{t}{t_0}\right)^{-\beta} \frac{r}{R_0} \right]^{\frac{n+1}{n}} \right\}^{\frac{n}{2n+1}},$$

(12)

$$\alpha = \frac{2 - (n+1)\lambda}{5n+3}, \qquad \beta = \frac{1 + (2n+1)\lambda}{5n+3}, \qquad t_0 = \frac{\beta}{\Gamma}\left(\frac{2n+1}{n+1}\right)^n \frac{R_0^{n+1}}{H_0^{2n+1}}, \qquad \Gamma = \frac{2A}{5}(\rho g)^n.$$

(13)

The special case of zero mass balance, described by $\lambda = 0$, gives the solution by Halfar (1981). The results of this experiment are shown in Fig. 4, for the case of $\lambda = 5$, which describes a positive accumulation rate, resulting in a rapidly expanding ice sheet. Here, too, UFEMISM reproduces the analytical solution well, with the largest errors occurring at the ice-sheet margin, and decreasing with resolution.

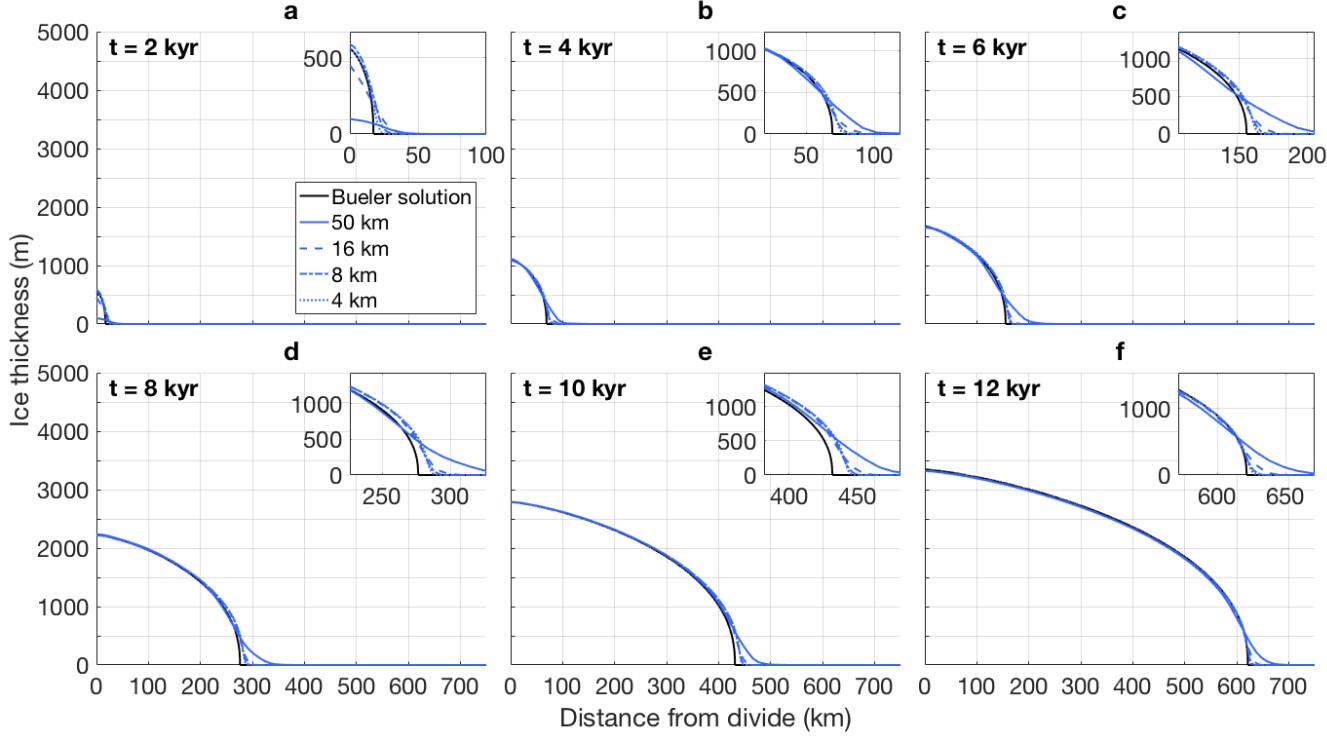

**Figure 4: the evolution through time of a schematic, radially symmetric, isothermal ice sheet, with positive accumulation rate, as simulated by UFEMISM at different ice-margin resolutions, compared to the Bueler (2005) solution.**

### 3.2 EISMINT benchmark experiments

In order to further investigate the validity of our numerical schemes for ice dynamics and thermodynamics, we used UFEMISM to perform the first set of EISMINT benchmark experiments (Huybrechts et al., 1996). All of the 6 experiments consist of a radially symmetric ice sheet, lying atop a flat, non-deformable bedrock. While the temperature of the ice is calculated dynamically, the ice flow factor is kept fixed at a value of $A = 10^{-16}$ Pa$^3$ yr$^{-1}$, meaning that ice temperature is a purely diagnostic variable. The six experiments are divided into two groups of three: a "moving margin" and a "fixed margin" group; in the "fixed margin" experiments, the mass balance is such that the expected theoretical ice margin lies outside the model domain, and ice thickness at the boundary is artificially kept at zero. In realistic model configurations, such a margin (i.e. where the ice thickness does not approach zero) can occur at a calving front. A moving margin is achieved by setting a zero mass balance integral over a bounded region fully enclosed within the model domain. The three experiments within a group have different mass balances; a fixed, "steady-state" mass balance, one with an added 20 kyr sinusoid, and one with a 40 kyr sinusoid, which is useful for investigating the performance of the model in terms of temporal evolution. Simulations for each experiment were performed with ice-margin resolutions of the original EISMINT resolution of 50 km, as well as finer resolutions of 16, 8 and 4 km.

**Table 1: The six different EISMINT experiments.**

| Experiment | Margin | Mass Balance |
|---|---|---|
| I | moving | steady-state |
| II | moving | 20 kyr |
| III | moving | 40 kyr |
| IV | fixed | steady-state |
| V | fixed | 20 kyr |
| VI | fixed | 40 kyr |

Shown in Fig. 5 are the simulated ice thickness and ice velocity of a radial transect of the ice sheet in experiment I, at the end of a 120 kyr simulation that was initialised with an ice thickness of zero. These results agree well with those presented by Huybrechts et al. (1996), showing an ice sheet that is ~2960 m thick at the divide, and has a maximum outward ice velocity of ~55 m yr$^{-1}$ at approximately 450 km away from the divide. The small sub-panel in the upper right corner of panel A zooms in on the ice margin, showing that the modelled ice margin converges to the analytical ice margin (the perimeter of the circle where the mass balance integrates to zero) as the resolution increases.

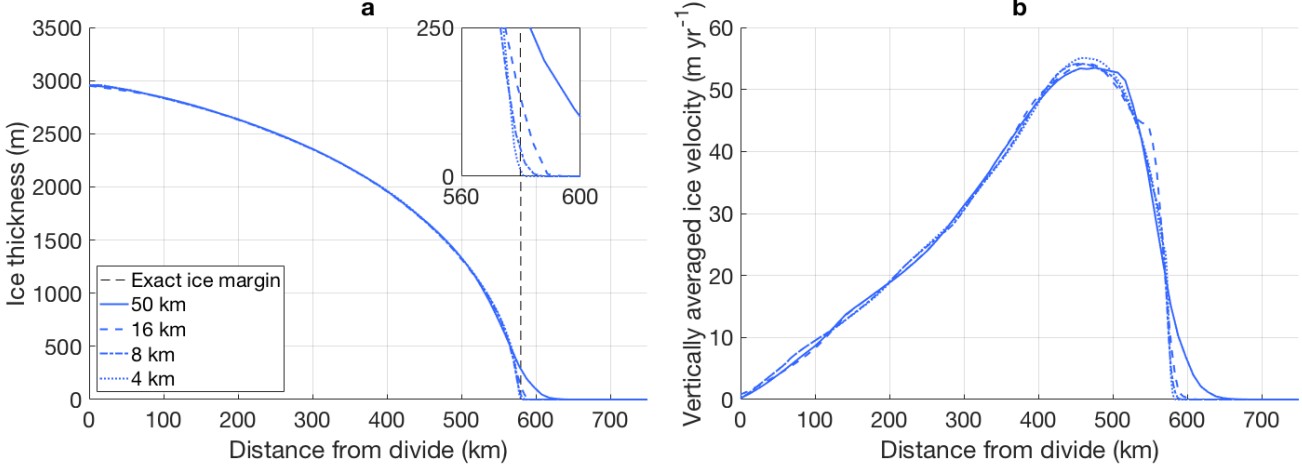

**Figure 5: a) Ice thickness and b) vertically averaged ice velocity of a radial transect of the ice sheet in experiment I, simulated by UFEMISM at 50, 16, 8, and 4 km ice-margin resolutions. The vertical dashed line in panel a denotes the analytical ice margin.**

Shown in Fig. 6 are the same transects for experiment IV (steady-state, fixed margin), showing an ice sheet that is ~3400 m thick at the divide, compared to 3340 – 3420 m in Huybrechts et al. (1996).

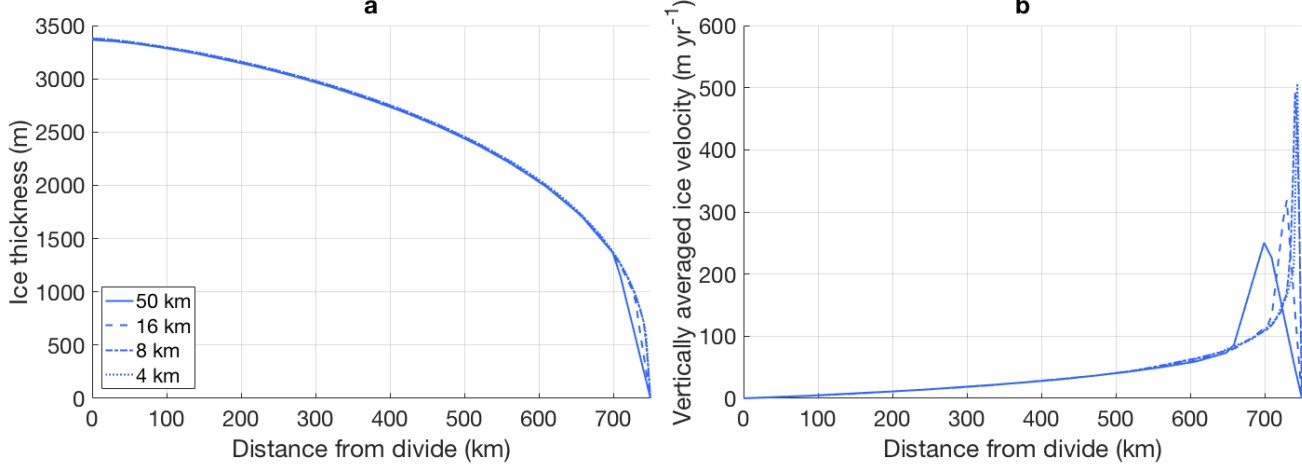

**Figure 6: a) Ice thickness and b) vertically averaged ice velocity of a radial transect of the ice sheet in experiment IV, simulated by UFEMISM at 50, 16, 8, and 4 km ice-margin resolutions. The sharp peaks in the velocity near the margin are a result of the singularity in these Nye-Vialov margins, the ice thickness approaches zero as one approaches the margin, but the ice flux remains finite, implying an infinite velocity.**

Figs. 7 and 8 show the temperature transect of the ice sheet for experiments I and IV at 4 km resolution, and the basal temperature transects for all simulations in these experiments. Following the specifications from Huybrechts at al. (1996), the thermal conductivity and specific heat capacity of ice are kept fixed at uniform values of $k = 2.1$ J s$^{-1}$ m$^{-1}$ K$^{-1}$ and $c_p = 2009$ J kg$^{-1}$ K$^{-1}$, respectively. Here, too, results agree well to those reported by Huybrechts et al. (1996). In experiment I, basal temperatures at the ice divide are $11 - 12$ K below the pressure melting point (PMP), increasing along the outward transect until they reach the PMP at $350 - 400$ km from the divide. In experiment IV we see similar results, with basal temperatures at the ice divide lying around 8 K below PMP, reaching PMP slightly further towards the margin at $400 - 450$ km from the divide. Preliminary experiments with a one-dimension set-up (vertical column only) show that these results are robust across different choices of vertical resolutions.

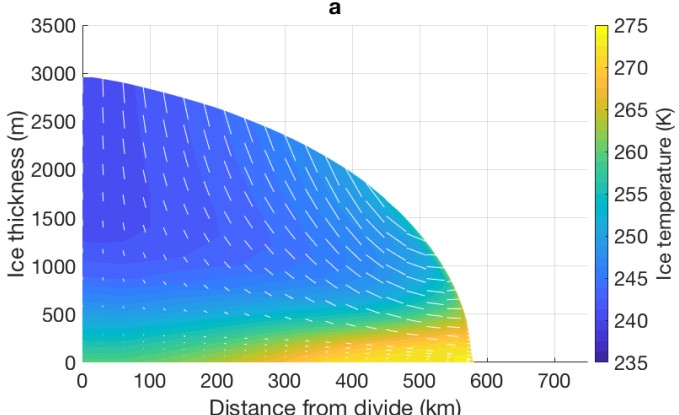 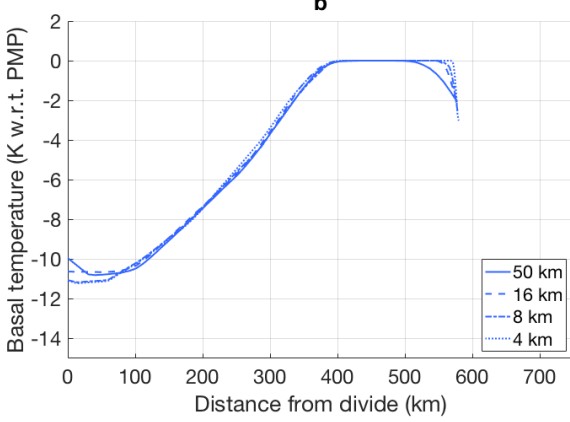

**Figure 7: a) Ice temperature and velocity vectors for the steady-state ice sheet in experiment I, as produced by UFEMISM with a 4 km ice margin resolution. b) Basal temperature transects for the different simulations in the same experiment.**

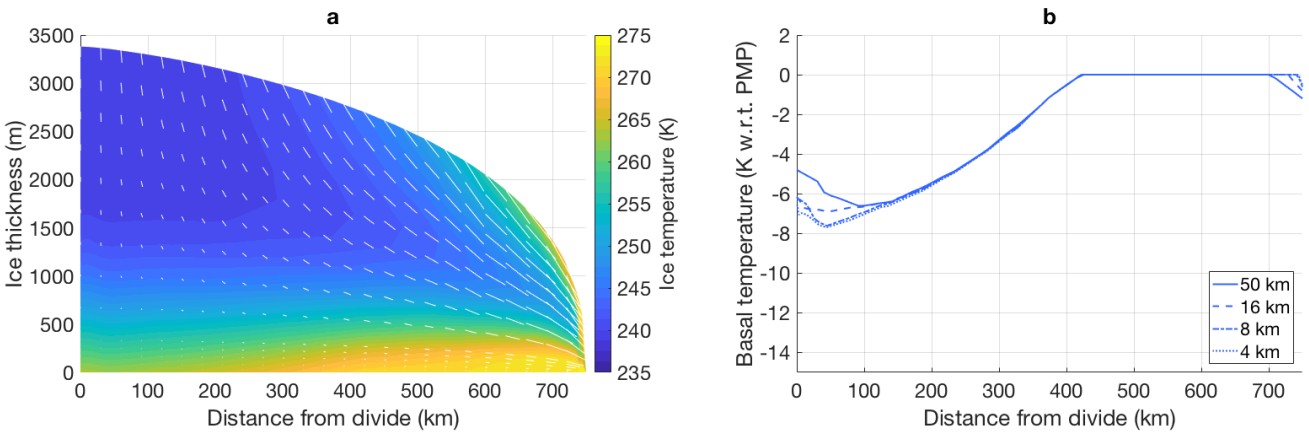

**Figure 8: a) Ice temperature and velocity vectors for the steady-state ice sheet in experiment IV, as produced by UFEMISM with a 4 km ice margin resolution. b) Basal temperature transects for the different simulations in the same experiment.**

Fig. 9 shows the ice thickness change at the ice divide over time for experiments II, III, V and VI. Here too, the results from simulations with UFEMISM at different resolutions agree with the results published by Huybrechts et al. (1996). The glacial-interglacial ice thickness (G-IG) changes for all four experiments lie within the ranges reported by Huybrechts et al. (1996), as listed in the top-right corners of both panels of Fig. 9. The simulated ice thickness timeseries at different resolutions are not distinguishable. Fig. 10 shows the basal temperature relative to the pressure melting point over time, for the same experiments. We find G-IG temperature changes that are slightly smaller than the values reported by Huybrechts et al. (1996), lying just outside their reported ranges. In agreement with the findings of Huybrechts et al. (1996), introducing glacial cycles in the moving margin experiments (Fig. 10a) results in G-IG mean temperature decrease of about 1 K, while the fixed margin experiments (Fig. 10b) see a temperature increase of about 2.5 K.

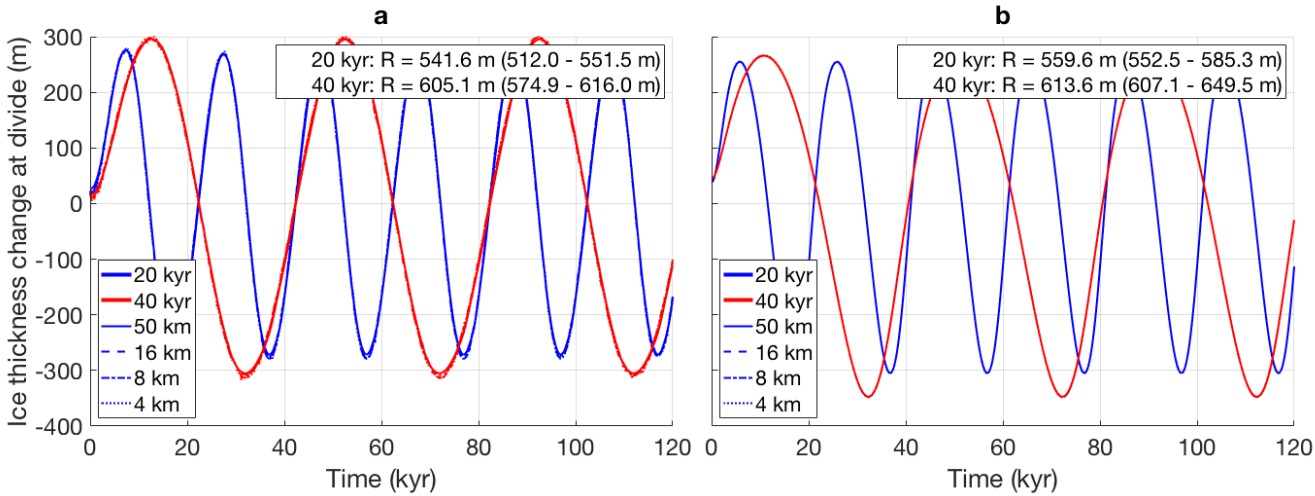

**Figure 9: a) Ice thickness change at the divide for experiments II and III (moving margin, respectively 20 kyr (blue) and 40 kyr (red) sinusoid mass balance perturbation). B) The same for the fixed margin experiments (V and VI), simulated by UFEMISM at 50, 16, 8, and 4 km ice-margin resolutions. Listed in the top-right corners of both panels are the glacial-interglacial ice thickness changes simulated by UFEMISM, compared to the ranges reported by Huybrechts et al. (1996) between brackets.**

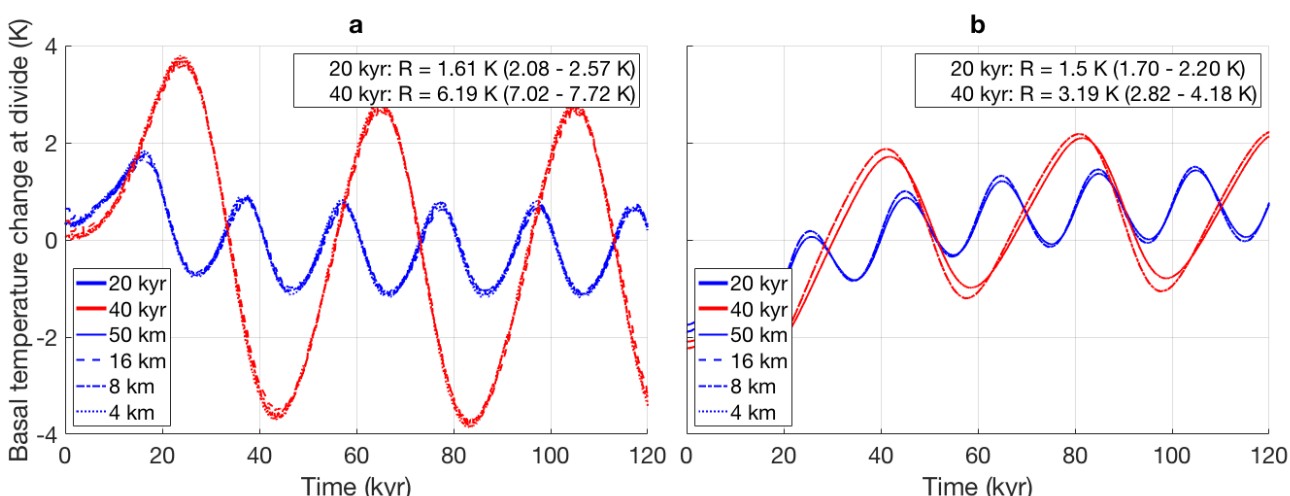

**Figure 10: a) Basal temperature change at the divide for experiments II and III (fixed margin, respectively 20 kyr (blue) and 40 kyr (red) sinusoid mass balance perturbation). b) The same for the fixed margin experiments (V and VI), simulated by UFEMISM at 50, 16, 8, and 4 km ice-margin resolutions. Listed in the top-right corners of both panels are the glacial-interglacial basal temperature changes simulated by UFEMISM, compared to the ranges reported by Huybrechts et al. (1996) between brackets.**

## 3.3 MISMIP benchmark experiment

In order to validate our solution to the SSA, we performed the first MISMIP experiment (Pattyn et al., 2012), modified from a 1-D flowline experiment to 2-D plan-view experiment in a manner similar to Pattyn (2017). This schematic experiment consists of a cone-shaped island at the centre of the model grid (described by $b = 720 - 778.5 \frac{\sqrt{x^2+y^2}}{750 \ km}$), under a spatially and temporally uniform mass balance forcing of 30 cm yr$^{-1}$. This results in a circular ice sheet, surrounded by an ice shelf that

extends to the domain boundary. In order to assess grounding line dynamics in the model, the ice flow factor $A$ is step-wise increased (leading to grounding-line advance) and subsequently decreased (leading to grounding-line retreat). Pattyn et al. (2012) showed that, while most participating ice-sheet models produce some amount of grounding line advance in the first phase, many of them failed to retreat back to their initial position in the second phase. The "best" performance (i.e. a one-to-one relation between flow factor and grounding-line position, without any hysteresis) was observed in models that included a

semi-analytical solution to the grounding line flux (Schoof, 2007; Tsai et al., 2012), either as a boundary condition to the SSA, or by overwriting the numerically derived grounding line flux (typically using a heuristic rule to determine which pixels to apply the analytical solution to). We chose for the former approach in UFEMISM, using the semi-analytical solution by Tsai et al. (2012) for a grounding line flux with a Coulomb-type sliding law, as a boundary condition in the SSA. This was achieved by solving the SSA simultaneously on both the regular and the "staggered" mesh (explained in more detail in Appendix C),

and keeping the values on the staggered grounding line vertices (lying halfway between a grounded and a floating vertex) fixed at the analytical solution. We then prescribe a flow factor with step-wise changes every 25,000 years, as shown in Table 2. This experiment was performed with grounding-line resolutions of 64 km, 32 km, and 16 km.

**Table 2: The step-wise flow factor changes in the MISMIP experiment.**

| Time window | Flow factor ($Pa^{-3}$ $yr^{-1}$) |
|---|---|
| $0 - 25$ kyr | $10^{-16}$ |
| $25 - 50$ kyr | $10^{-17}$ |
| $50 - 75$ kyr | $10^{-16}$ |

The results of this experiment with UFEMISM are shown in Fig. 11. Panel A shows cross-sections of the modelled ice sheets at the end of the three time windows. As can be seen, the ice sheets at 25 kyr and 75 kyr are nearly indistinguishable (as they should be for the same flow factor), and no appreciable dependence on resolution. Panel B shows the grounding line position over time, showing that the grounding line retreats exactly to its initial position after the flow factor is returned to its initial value, and that the results are resolution-independent.

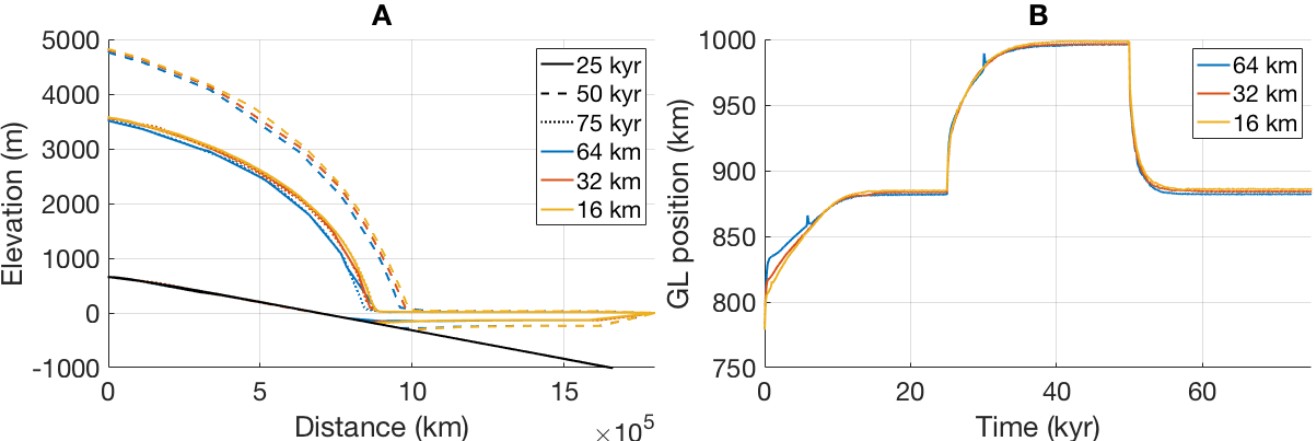

Figure 11: a) Cross-sections of the modelled ice sheet at different times and different resolutions. b) Grounding-line position over time for the different resolutions.

# 4 Computational performance

The first version of UFEMISM presented here is parallelised using the Message Passing Interface (MPI) construct of "shared memory", allowing the program to run in parallel on a number of processor cores that are able to directly access the same physical memory (usually either 16, 24, or 32 cores on typical computer clusters or supercomputers). This is a temporary

choice; the effort to extend the parallelisation to multiple nodes, using the full capability of MPI, is still ongoing, but is beyond the scope of this paper.

In order to investigate the computational performance of the different model components, we performed a series of simulations of Antarctic ice-sheet retreat over a period of 10,000 years. For these simulations, the climate was set to the present-day observed climate (ERA-40; Uppala et al., 2005) plus a spatially and temporally uniform 5 °C warming. The version of UFEMISM presented here already includes the IMAU-ITM mass balance model (Fettweis et al., 2020), so that a 5 °C warming leads to a substantial retreat through the increase in surface melt. This experiment is not meant to represent a realistic projection of Antarctic retreat; the choice of forcing is merely convenient because it ensures a rapidly changing ice-sheet geometry, forcing frequent mesh updates. The results of one of these simulations are shown in Fig. 12.

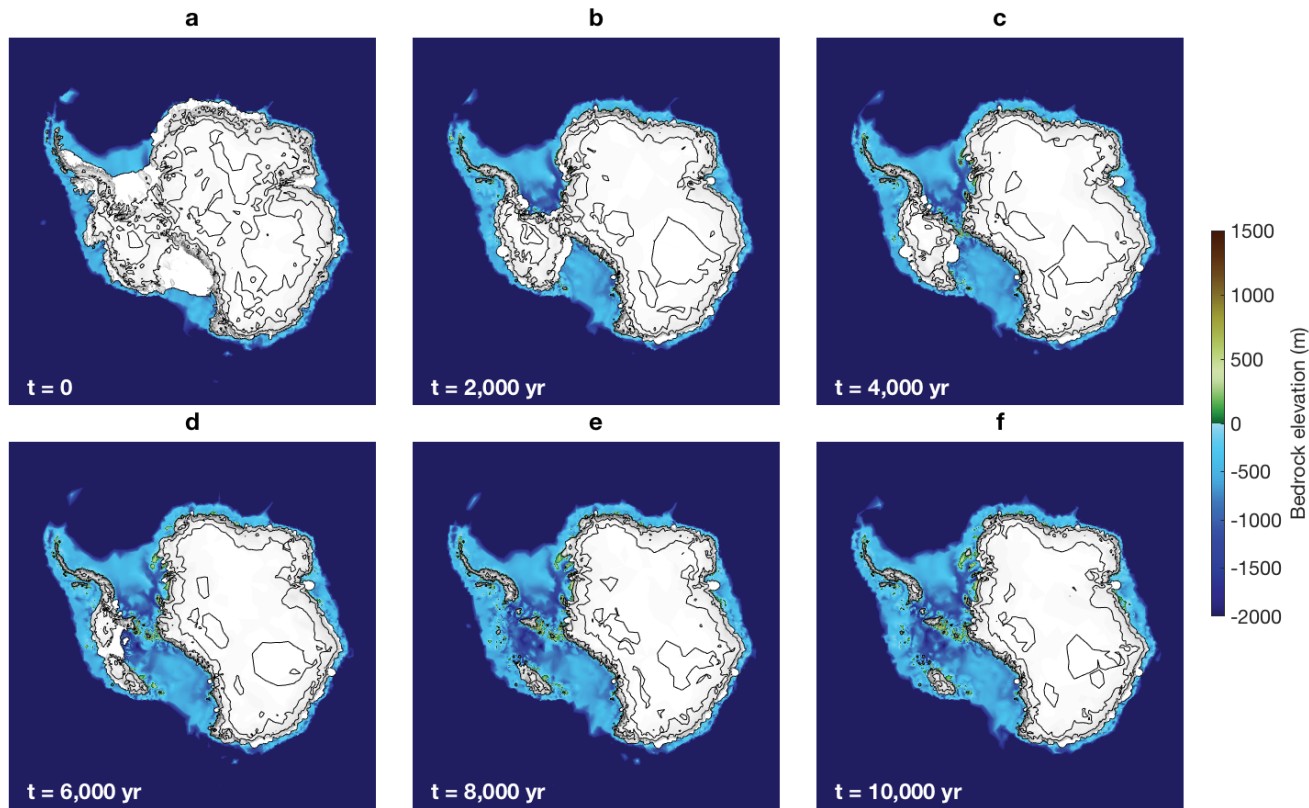

**Figure 12: Results from the 10,000 yr Antarctic retreat simulation with a 4 km grounding line resolution. Panels a-f show the modelled ice sheet at 2,000 yr intervals. Bedrock elevation is indicated by colours, surface elevation contours are shown at 1,000 m intervals.**

The computation times of the different model components as a function of number of cores and model resolution are shown in Fig. 13. The simulations described in Sect. 4 were run on the LISA computer cluster operated by SURFsara, using an Intel Xeon Gold 6130 Processor with a 2.1 GHz clock frequency and 22 MB cache, and were compiled with the iFort compiler.

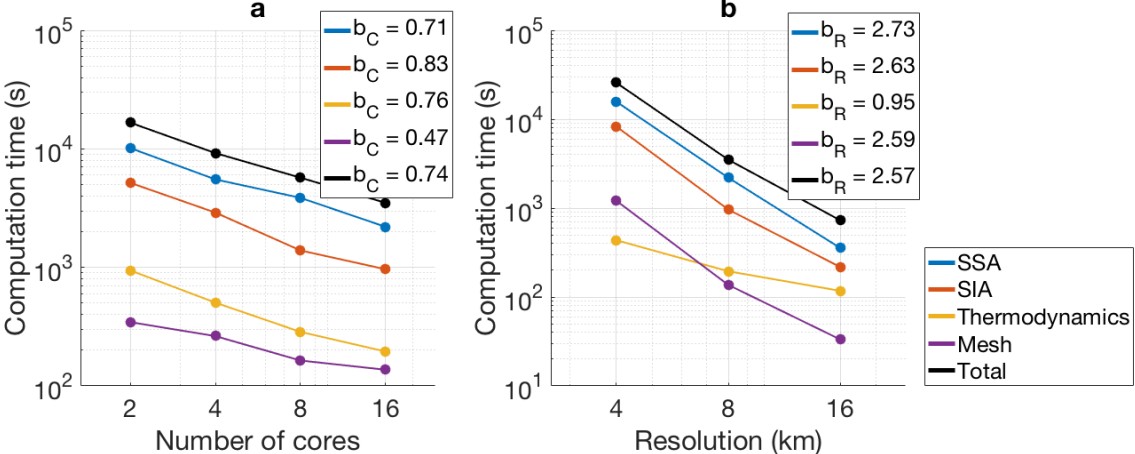

**Figure 13: a)** Computation times vs number of cores for the SSA (blue), SIA (red), thermodynamics (yellow) and mesh updating (purple) model components, as well as for the entire model (black), for a 10,000 yr Antarctic retreat simulation at 8 km resolution. These four components together account for ~99.8 % of the total computation time. Logarithmic fits of the form $\ln t = a + b_C \ln C$ have been made to the data, with the scaling coefficients $b_C$ shown in the legend. **b)** Computation times vs model resolution for the same model components, run on 16 cores. Logarithmic fits of the form $\ln t = a + b_R \ln R$ have been made to the data, with the scaling coefficients $b_C$ shown in the legend.

Fig. 13A shows the degree of parallelisation of the model components. For each model component, logarithmic fits have been made between the computation time $t$ and the number of cores $C$ of the form $\ln t = a + b_C \ln C$. The coefficient $b_C$ describes the degree of parallelisation, such that $b_C = 1$ describes perfect parallelisation, i.e. doubling the number of cores reduces the computation time by half. The three physics modules have a good degree of parallelisation, ranging from $b_C = 0.71$ for the SSA to $b_C = 0.83$ for the SIA. Mesh updating scales less well, with $b_C = 0.47$. This has likely to do with the fact that the entire mesh generation code was writing by the authors themselves, instead of relying upon available external software packages. Since mesh updating accounts for only $2 - 4$ % of total computation time, this does not noticeably affect the parallelisation of the complete model, which has $b_C = 0.74$. The SSA dominates the total computation time across all resolutions and numbers of processors, requiring as much or more computation time as all other model components combined. The routine that applies the finite volume method described in Appendix B to update the ice thickness through time is included in the SIA computation time.

Fig. 13 B shows computation versus model resolution for the same model components. For each model component, logarithmic fits have been made between the computation time $t$ and the resolution $R$ of the form $\ln t = a + b_R \ln R$. In an idealised situation (such as the EISMINT experiments), the SIA and SSA should scale with the resolution to order $b_R = 3$ in a square-grid model (two orders from the number of grid cells, and one from the time-step dependence on resolution, according to the numerical stability criterion for the solution of the mass conservation equation). In UFEMISM, this can be theoretically reduced to order $b_R = 2$, since a high resolution is only applied over the one-dimensional ice-sheet margin, where the diffusivity is not necessarily the largest of the model domain. The reason that the results shown in Fig. 13 deviate from this idealised case is

because, for a realistic ice sheet, increasing the resolution resolves more topographical features, which increases the length of the one-dimensional domains of the ice margin, the grounding line, and the calving front. This is illustrated in Fig. 14, which shows the number of vertices of meshes for Greenland and Antarctica versus the model resolution at the ice margin.

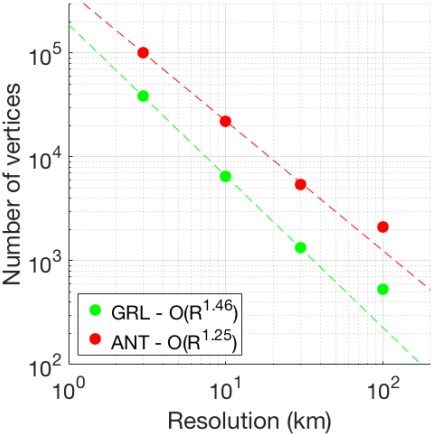

Figure 14: **Number of vertices vs ice margin resolution for Greenland and Antarctica.**

Similarly, resolving more topographical features also leads to an increased ice diffusivity in areas with steep gradients, decreasing the critical time step and increasing the computation time. Preliminary experiments with a simply square-grid model showed the same effect, and the computation time for those experiments scaled with resolution to order $b_R = 3.5 - 4.0$, rather than the $b_R = 3$ in the idealised case.

The computation time of the mesh updating component also scales well with model resolution, to the order $b_R = 2.59$. The thermodynamics component scales even better, with order $b_R = 0.95$. The reason that this is so much lower than the SIA and SSA components, is that the thermodynamics uses a constant rather than a dynamic time step, which is related to the choice of vertical discretisation rather than to the horizontal model resolution.

In these experiments, the entire ice margin, including the grounding line and calving front, was given the same resolution. When using UFEMISM for actual paleo-simulations, such a high resolution would only be used for the grounding line, where it directly increases the physical accuracy of the solution. In one last experiment, we performed the same Antarctic retreat simulation, with the resolution set to 4 km for the grounding line, 16 km for the rest of the ice margin, and 200 km over the ice sheet interior. Run on 16 cores, the simulation took about 88 core hours (5 h 30 m wall-clock time) to complete. Extrapolating these numbers to the Greenland, North American and Eurasian ice sheets is difficult, due to the differences in size, glacial-interglacial geometry changes, and relative area of floating ice. Previous work with the square-grid model ANICE (Berends et al., 2018, 2019) indicates that the Antarctic and North American ice sheets typically have roughly the same computational expense when they are at their maximum extent, while the Eurasian and Greenland ice sheets respectively

require about ½ and ¼ as much computation time. Based on these numbers, a full glacial cycle simulation of all four ice sheets would take somewhere between 1.5 and 3 times as much as one for only Antarctica, implying a required computation time of about 1,600 – 3,200 core hours (100 – 200 wall clock hours on a 16-core system). For comparison, a full glacial cycle simulation with ANICE at 40 km resolution takes about 8 core hours, which implies a simulation at 4 km would take about 25,000 – 80,000 core hours (numbers based on extrapolation), meaning that UFEMISM is about 10 – 30 times faster. If the grounding line resolution is decreased to 8 km, these numbers decrease to about 200 – 600 core hours (20 – 40 wall hours) for UFEMISM, and 2,200 – 5000 core hours for ANICE. These numbers are summarised in Table 3.

**Table 3: Observed and extrapolated computation times (in core hours) for different simulations with ANICE and UFEMISM. Bold-faced numbers are observed times, all others are estimates based on extrapolation.**

| Experiment | ANICE | UFEMISM |
| --- | --- | --- |
| 10 kyr Antarctic retreat, 4 km | 700 – 4,500 | **88** |
| 120 kyr glacial cycle (all ice), 40 km | **8** | 2.5 – 12.5 |
| 120 kyr glacial cycle (all ice), 8 km | 2,200 – 5,000 | 200 – 600 |
| 120 kyr glacial cycle (all ice), 4 km | 25,000 – 80,000 | 1,600 – 3,200 |

UFEMISM can be compiled with both the gfortran and ifort compilers, requiring only LAPACK, NetCDF and MPI as external packages, and can run on any number of processors that can access the same shared memory chip. This means that it should be possible to compile and run the model on most consumer-grade systems. The model contains roughly 200 double precision data fields. For the 3 km resolution Antarctica mesh (~100,000 vertices) shown in Fig. 1, this implies a memory usage of about 1.6 Gb. Output is written to NetCDF files at about 10 kb per vertex, implying that a 120 kyr glacial cycle simulation of Antarctica at 3 km (100,000 vertices), where output is written every 1,000 model years, would generate about 150 Gb of data.

**Conclusions and discussion**

We have presented and evaluated a new thermomechanically coupled ice-sheet model, UFEMISM, which solves the SIA and SSA versions of the ice-dynamical equations on a fully adaptive mesh. The model is able to accurately reproduce the analytical solutions for the evolution of schematic ice sheets by Halfar (1981) and Bueler et al. (2005), as well as the EISMINT benchmark experiments (Huybrechts et al., 1996), indicating that the numerical schemes for solving the SIA and integrating the mass conservation law are valid. In a modified version of the first MISMIP experiment (Pattyn et al., 2012), adapted from a 1-D flowline to a 2-D plan view, UFEMISM shows a grounding line position that is resolution-independent, and displays no hysteresis during forced advance-retreat cycles. Analysis of the computational performance of the model indicates that it scales well with both number of processors and model resolution. Based on those results, UFEMISM should be able to simulate the evolution of the four large continental ice sheets over an entire glacial cycle, with a grounding line resolution of 4 km, in 1600

– 3200 core hours (100 – 200 wall hours on 16 processors), which is very feasible on typical facilities for scientific computation.

The MISMIP experiment (Pattyn et al., 2012) used here to validate our solution to the SSA requires further consideration. Pattyn et al. (2012) showed that simply solving the SSA without any special treatment of the grounding line results in grounding line positions that are strongly resolution-dependent, and yield significant hysteresis during advance-retreat cycles, unless model resolution is lower than ~100 m (a value that is not feasible in palaeoglaciological simulations). Different semi-analytical solutions for the ice flux across the grounding line in the absence of buttressing have been proposed (Schoof et al., 2007; Tsai et al., 2015), and Pattyn et al. (2012) showed that using these solutions as a boundary condition decreases both the resolution dependence and the hysteresis. However, implementing these analytical solutions in 3D numerical models has proven difficult (Pollard and DeConto, 2012; Pattyn et al., 2013), and recent work has demonstrated that these analytical solutions cannot be feasibly altered to account for buttressing (Reese et al., 2018), which is absent in the MISMIP experiment we performed, but which plays an important role at the majority of Antarctic grounding lines (Reese et al., 2018). Our model therefore meets the same performance standards as existing models for grounding line dynamics, but the schematic test itself are not a sufficient condition for correct grounding line dynamics under all circumstances. The role of buttressing remains a matter of debate. It is also important to mention that similar problems (resolution dependence and hysteresis) are observed in full-Stokes models (Gagliardini et al., 2016), which suggests that these problems are not specific to the SSA.

The current version of UFEMISM has been parallelised using MPI shared memory, meaning the number of processors that can run the model depends on the hardware system, limited by how many processors can access the same memory chip. The three most computationally expensive model components (the SSA, SIA and thermodynamics modules, respectively) are shown to scale well. Extending the parallelisation framework to allow the model to run on multiple nodes might therefore substantially reduce the wall clock time for large simulations. The current version of UFEMISM uses domain decomposition to parallelise the mesh generation algorithm. This approach lends itself well to multi-node parallelisation, as each node can be assigned a region of the model domain.

The iterative scheme used in solving the SSA is currently the most computationally demanding model component, requiring as much or more computation time as all the other model components combined. Preliminary experiments have identified several possible solutions that could substantially reduce this, including a spatially variable relaxation parameter (see Appendix B) and/or a multigrid scheme. Efforts to develop these solutions into applications that are robust and stable enough for the large-scale, long-term simulations for which UFEMISM is intended are ongoing, but are beyond the scope of this study.

As UFEMISM is intended to be used for paleo-simulations, it has been developed to be able to include an elaborate climate / mass balance forcing. Previous work by our group has focused on developing a computationally efficient climate forcing that

explicitly includes important feedback processes in the ice-climate system, such as the altitude-temperature feedback, ice-albedo feedback and orographic precipitation feedback (Berends et al., 2018). While these solutions have not yet been implemented in UFEMISM, it has been designed with these solutions in mind, so that implementing them should be straightforward. Future work will also focus on improving the thermodynamics (probably using an energy-conserving

enthalpy-based approach along the lines of Aschwanden et al., 2012), adding a basal hydrology model (Bueler and van Pelt, 2015), and including recently developed parameterisations for cliff failure and shelf hydrofracturing (Pollard et al., 2015). We have also previously worked on developing a coupled ice-sheet – sea-level model (de Boer et al., 2014). A key improvement in UFEMISM with respect to our previous ice-sheet model is the improved treatment of grounding line dynamics, which makes it important to accurately account for changes in the geoid and the resulting changes in water depth at the grounding line.

While UFEMISM has not yet been coupled to a sea-level model, it has been designed with such a future coupling in mind.

**Code and data availability**

The Fortran90 source code of UFEMISM, as well as a collection of Matlab scripts for analysing and visualising output data, and some more elaborate documentation, are freely available at http://doi.org/10.5281/zenodo.4001592. The different benchmark experiments described in Sect. 3 can be run with the supplied config files. The Antarctic retreat simulations

described in Sect. 4 require input files describing the present-day Antarctic ice sheet and climate. For the ice-sheet and bedrock geometry, we used the Bedmachine Antarctica v1 data by Morlighem et al. (2019) which is freely available as supplementary material to their publication. For the present-day climate we used the ERA-40 reanalysis (Uppala et al., 2005), which can be downloaded from the ECMWF website (https://www.ecmwf.int/) after creating a free user account.

*Author contributions*. CJB created the model and performed the experiments, with some helpful input from HG. CJB drafted the first version of the manuscript, all authors contributed to the final version.

*Competing interests*. Heiko Goelzer is a member of the editorial board of the journal.

*Financial support*. This publication was generated in the frame of Beyond EPICA. The project has received funding from the European Union's Horizon 2020 research and innovation programme under grant agreement No. 815384 (Oldest Ice Core). It is supported by national partners and funding agencies in Belgium, Denmark, France, Germany, Italy, Norway, Sweden, Switzerland, The Netherlands and the United Kingdom. Logistic support is mainly provided by PNRA and IPEV through the Concordia Station system. HG was funded by the program of the Netherlands Earth System Science Centre (NESSC; Dutch

Ministry of Education, Culture and Science [OCW] grant no. 024.002.001). The use of supercomputer facilities was sponsored by NWO Exact and Natural Sciences. Model runs were performed on the LISA Computer Cluster, we would like to acknowledge SurfSARA Computing and Networking Services for their support. The opinions expressed and arguments

employed herein do not necessarily reflect the official views of the European Union funding agency or other national funding bodies. This is Beyond EPICA publication number XX.

## Appendix A – Discretising derivatives on an unstructured triangular mesh

### A.1 First-order partial derivatives

In UFEMISM, derivatives are discretised on the unstructured triangular mesh using an averaged-gradients approach. In short, gradients are defined on the triangles surrounding a vertex (which are piecewise smooth surfaces, having unique gradients). The gradient on a vertex is then defined simply as the average of the gradients on the surrounding triangles. In the following derivation, we will show that this results in a linear combination of the function values on a vertex and its direct neighbours. These coefficients, which we here call "neighbour functions" (also known as "numerical stencils"), are a function of mesh

geometry, which means they need to be calculated only once for a new mesh, and can be stored in memory for later use. This averaged-gradient approach is very similar to an unweighted least-squares approach (Syrakos et al., 2017), and in Sect. A.3 we will show that, as expected, the resulting accuracy is 2nd order convergent for the first derivatives $f_x$ and $f_y$, and 1st order convergent for the second derivatives $f_{xx}$, $f_{xy}$ and $f_{yy}$.

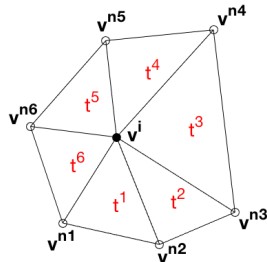

**Figure A1: a very simple unstructured triangular mesh. The vertex under consideration, $v^i$, is surrounded by its neighbours $v^{n1}$ to $v^{n6}$, which together span triangles $t^1$ to $t^6$. All neighbouring vertices and triangles are ordered counter-clockwise.**

Before getting started, we introduce "star notation" as shorthand for the modulo function: $i^* = \mathrm{mod}(i, n)$. Using this notation, "the next neighbour vertex counter-clockwise from neighbour $s$" becomes $(s + 1)^*$, while "the next surrounding triangle clockwise from triangle $t$" becomes $(t - 1)^*$.

Consider the unstructured triangular mesh in Fig. A1. We first define the partial derivatives $f_x$ and $f_y$ of a function $f$ on the triangles surrounding vertex $i$. Since these triangles are plane sections, they have well-defined first-order derivatives, which can be expressed using the normal vector to the plane. Treating the value $f^i$ of $f$ on vertex $i$ as a coordinate in a third dimension, we can define $\boldsymbol{v}^i = [x^i, y^i, f^i]$, such that the upward normal vector $\boldsymbol{n}^t$ to triangle $t$, spanned by $\boldsymbol{v}^i$, $\boldsymbol{v}^t$ and $\boldsymbol{v}^{(t+1)^*}$,

is given by:

$$\boldsymbol{n}^t = (\boldsymbol{v}^t - \boldsymbol{v}^i) \times (\boldsymbol{v}^{(t+1)^*} - \boldsymbol{v}^i) = \begin{bmatrix} f^i\left(y^{(t+1)^*} - y^t\right) + f^t\left(y^i - y^{(t+1)^*}\right) + f^{(t+1)^*}\left(y^t - y^i\right) \\ f^i\left(x^t - x^{(t+1)^*}\right) + f^t\left(x^{(t+1)^*} - x^i\right) + f^{(t+1)^*}\left(x^i - x^t\right) \\ \left(x^t - x^i\right)\left(y^{(t+1)^*} - y^i\right) - \left(y^t - y^i\right)\left(x^{(t+1)^*} - x^i\right) \end{bmatrix}. \quad \text{(A1)}$$

The first-order spatial derivatives $f^t_{x,tri}$ and $f^t_{y,tri}$ of $f$ on $t$ are then given by:

$$f^t_{x,tri} = \frac{-n^t_x}{n^t_z}, f^t_y = \frac{-n^t_y}{n^t_z}. \quad \text{(A2)}$$

These expressions can be simplified by introducing the "neighbour functions" on triangle $t$:

$$N^t_{x,tri} = \frac{1}{n^t_z}\left[y^t - y^{(t+1)^*}, y^{(t+1)^*} - y^i, y^i - y^t\right], \quad \text{(A3a)}$$

$$N^t_{y,tri} = \frac{1}{n^t_z}\left[x^{(t+1)^*} - x^t, x^i - x^{(t+1)^*}, x^t - x^i\right]. \quad \text{(A3b)}$$

Eqs. A2 can then be written as:

$$f^t_{x,tri} = f^i N^t_{x,tri}(1) + f^t N^t_{x,tri}(2) + f^{(t+1)^*} N^t_{x,tri}(3), \quad \text{(A4a)}$$

$$f^t_{y,tri} = f^i N^t_{y,tri}(1) + f^t N^t_{y,tri}(2) + f^{(t+1)^*} N^t_{y,tri}(3). \quad \text{(A4b)}$$

We approximate the derivative $f^i_x$ of $f$ on vertex $i$ by averaging the derivatives on the $n$ surrounding triangles:

$$f^i_x = \frac{1}{n}\sum_{t=1}^{n} f^t_{x,tri} = \frac{1}{n}\sum_{t=1}^{n}\left[f^i N^t_{x,tri}(1) + f^t N^t_{x,tri}(2) + f^{(t+1)^*} N^t_{x,tri}(3)\right] \quad \text{(A5)}$$

$$= f^i \frac{1}{n}\sum_{t=1}^{n}\left[N^t_{x,tri}(1)\right] + \frac{1}{n}\sum_{t=1}^{n}\left[f^t N^t_{x,tri}(2)\right] + \frac{1}{n}\sum_{t=1}^{n}\left[f^{(t+1)^*} N^t_{x,tri}(3)\right].$$

5   Lowering the indices in the last sum term by one, this can be rearranged to read:

$$f^i_x = \frac{1}{n}\sum_{t=1}^{n} f^t_{x,tri} = f^i \frac{1}{n}\sum_{t=1}^{n}\left[N^t_{x,tri}(1)\right] + \frac{1}{n}\sum_{t=1}^{n}\left[f^t N^t_{x,tri}(2)\right] + \frac{1}{n}\sum_{t=1}^{n}\left[f^t N^{(t-1)^*}_{x,tri}(3)\right] \quad \text{(A6)}$$

$$= f^i \frac{1}{n}\sum_{t=1}^{n}\left[N^t_{x,tri}(1)\right] + \frac{1}{n}\sum_{t=1}^{n}\left[f^t\left(N^t_{x,tri}(2) + N^{(t-1)^*}_{x,tri}(3)\right)\right].$$

This can be simplified by introducing the neighbour functions $N^i_x$ on vertex $i$:

$$N^i_x = \frac{1}{n}\sum_{t=1}^{n}\left[N^t_{x,tri}(1)\right], \quad \text{(A7a)}$$

$$N^{i,t}_x = \frac{1}{n}\left(N^t_{x,tri}(2) + N^{(t-1)^*}_{x,tri}(3)\right). \quad \text{(A7b)}$$

This simplifies Eq. A6 to:

$$f^i_x = f^i N^i_x + \sum_{t=1}^{n} f^t N^{i,t}_x. \quad \text{(A8)}$$

For vertices lying on the domain boundary, which therefore have $n$ neighbours but $n - 1$ surrounding triangles, it can be shown that the neighbour functions can be expressed as:

$$N_x^i = \frac{1}{n-1} \sum_{t=1}^{n-1} \left[ N_{x,tri}^t(1) \right], \tag{A9a}$$

$$N_x^{i,t} = \begin{cases} \dfrac{1}{n-1} N_{x,tri}^1(2) & \text{if } t = 1, \\[2mm] \dfrac{1}{n-1} N_{x,tri}^{n-1}(3) & \text{if } t = n, \\[2mm] \dfrac{1}{n-1} \left( N_{x,tri}^t(2) + N_{x,tri}^{(t-1)^*}(3) \right) & \text{otherwise.} \end{cases} \tag{A9b}$$

## A.2 Second-order partial derivatives

In order to discretise the second derivatives $f_{xx}$, $f_{xy}$ and $f_{yy}$, we treat the geometric centres of the triangles surrounding vertex $i$ as "staggered" vertices, where the staggered first derivatives are calculated according to Eqs. A4. We then construct a new set of "sub-triangles", spanned by the vertex $i$ and these staggered vertices, as illustrated in Fig. A2. Since we know the values of $f_x$ and $f_y$ on all of them, we can use the same approach as before to calculate the gradients of the first derivatives (i.e. the second derivatives) on the sub-triangles, and average them to get the values on vertex $i$. Preliminary experiments showed that

using the geometric centre instead of the circumcentre yields more stable solutions when using these to solve differential equations. A mathematical proof of why this is the case might be interesting, but lies beyond the scope of this study.

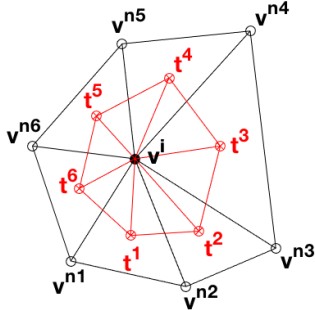

**Figure A2: the geometric centres of the triangles surrounding vertex $v^i$ make up the vertices of a new set of "sub-triangles", which can be used to define the second derivative of a function on vertex $v^i$.**

A new, staggered vertex in triangle $t$ is created, using the horizontal coordinates of the geometric centre of $t$, and the derivative $f_{x,tri}^t$ of $f$ on $t$ as the vertical coordinate:

$$v_x^t = \left[ \frac{x^i + x^t + x^{(t+1)^*}}{3}, \frac{y^i + y^t + y^{(t+1)^*}}{3}, f_{x,tri}^t \right]. \tag{A10}$$

The sub-triangle $s$ is spanned by $\boldsymbol{v}_x^i$, $\boldsymbol{v}_x^s$ and $\boldsymbol{v}_x^{(s+1)^*}$, where $\boldsymbol{v}_x^i = [x^i, y^i, f_x^i]$ (with $f_x^i$ as given by Eq. A8). The normal vector $\boldsymbol{n}_x^s$ to sub-triangle $s$ (using the derivatives of $f$ on the vertex $i$ and the triangles $s$ and $(s+1)^*$ as the vertical coordinate) is then given by:

$$\boldsymbol{n}_x^s = \left(\boldsymbol{v}_x^s - \boldsymbol{v}_x^i\right) \times \left(\boldsymbol{v}_x^{(s+1)^*} - \boldsymbol{v}_x^i\right)$$

$$= \frac{1}{3} \begin{bmatrix} f_x^i\left(y^{(s+2)^*} - y^s\right) + f_{x,tri}^s\left(2y^i - y^{(s+1)^*} - y^{(s+2)^*}\right) + f_{x,tri}^{(s+1)^*}\left(y^s + y^{(s+1)^*} - 2y^i\right) \\ f_x^i\left(x^s - x^{(s+2)^*}\right) + f_{x,tri}^s\left(x^{(s+1)^*} + x^{(s+2)^*} - 2x^i\right) + f_{x,tri}^{(s+1)^*}\left(2x^i - x^s - x^{(s+1)^*}\right) \\ \frac{1}{3}\left(x^s + x^{(s+1)^*} - 2x^i\right)\left(y^{(s+1)^*} + y^{(s+2)^*} - 2y^i\right) - \frac{1}{3}\left(y^s + y^{(s+1)^*} - 2y^i\right)\left(x^{(s+1)^*} + x^{(s+2)^*} - 2x^i\right) \end{bmatrix}. \quad \text{(A11)}$$

The second-order derivatives $f_{xx,sub}^s, f_{xy,sub}^s$ of $f$ on $s$ are then given by:

$$f_{xx,sub}^s = \frac{-n_{x,x}^s}{n_{x,z}^s}, \quad \text{(A12a)}$$

$$f_{xy,sub}^s = \frac{-n_{x,y}^s}{n_{x,z}^s}. \quad \text{(A12b)}$$

5 We then introduce the "sub-triangle neighbour functions", similar to Eqs. A3 (which, again, depend only on mesh geometry):

$$N_{x,sub}^s = \frac{1}{n_{x,z}^s}\left[\left(y^s - y^{(s+2)^*}\right), \left(y^{(s+1)^*} + y^{(s+2)^*} - 2y^i\right), \left(2y^i - y^s - y^{(s+1)^*}\right)\right], \quad \text{(A13a)}$$

$$N_{y,sub}^s = \frac{1}{n_{x,z}^s}\left[\left(x^{(s+2)^*} - x^s\right), \left(2x^i - x^{(s+1)^*} - x^{(s+2)^*}\right), \left(x^s + x^{(s+1)^*} - 2x^i\right)\right]. \quad \text{(A13b)}$$

Substituting these into Eqs. A12 yields:

$$f_{xx,sub}^s = f_x^i N_{x,sub}^s(1) + f_{x,tri}^s N_{x,sub}^s(2) + f_{x,tri}^{(s+1)^*} N_{x,sub}^s(3), \quad \text{(A14a)}$$

$$f_{xy,sub}^s = f_x^i N_{y,sub}^s(1) + f_{x,tri}^s N_{y,sub}^s(2) + f_{x,tri}^{(s+1)^*} N_{y,sub}^s(3). \quad \text{(A14b)}$$

We then substitute Eq. A4 and Eq. A8 into Eq. A14. From here on, only the xx-derivative is shown – xy and yy follow similar derivations:

$$f_{xx,sub}^s = N_{x,sub}^s(1)\left(f^i N_x^i + \sum_{t=1}^{n} f^t N_x^{i,t}\right)$$

$$+ N_{x,sub}^s(2)\left(f^i N_{x,tri}^s(1) + f^s N_{x,tri}^s(2) + f^{(s+1)^*} N_{x,tri}^s(3)\right)$$

$$+ N_{x,sub}^s(3)\left(f^i N_{x,tri}^{(s+1)^*}(1) + f^{(s+1)^*} N_{x,tri}^{(s+1)^*}(2) + f^{(s+2)^*} N_{x,tri}^{(s+1)^*}(3)\right) \quad \text{(A15)}$$

$$= f^i\left[N_{x,sub}^s(1)N_x^i + N_{x,sub}^s(2)N_{x,tri}^s(1) + N_{x,sub}^s(3)N_{x,tri}^{(s+1)^*}(1)\right] + f^s\left[N_{x,sub}^s(2)N_{x,tri}^s(2)\right]$$

$$+ f^{(s+1)^*}\left[N_{x,sub}^s(2)N_{x,tri}^s(3) + N_{x,sub}^s(3)N_{x,tri}^{(s+1)^*}(2)\right]$$

$$+ f^{(s+2)^*}\left[N_{x,sub}^s(3)N_{x,tri}^{(s+1)^*}(3)\right] + N_{x,sub}^s(1)\sum_{t=1}^{n} f^t N_x^{i,t}.$$

Again, we approximate the second derivative $f_{xx}^i$ of $f$ on $i$ by averaging over the surrounding sub-triangles:

$$f_{xx}^i = \frac{1}{n}\sum_{s=1}^{n} f_{xx,sub}^s$$

$$= f^i \frac{1}{n}\sum_{s=1}^{n}\left\{N_{x,sub}^s(1)N_x^i + N_{x,sub}^s(2)N_{x,tri}^s(1) + N_{x,sub}^s(3)N_{x,tri}^{(s+1)^*}(1)\right\}$$

$$+ \sum_{s=1}^{n}\left\{f^s\frac{1}{n}\left[N_{x,sub}^s(2)N_{x,tri}^s(2)\right]\right\}$$

$$+ \sum_{s=1}^{n}\left\{f^{(s+1)^*}\frac{1}{n}\left[N_{x,sub}^s(2)N_{x,tri}^s(3) + N_{x,sub}^s(3)N_{x,tri}^{(s+1)^*}(2)\right]\right\}$$ (A16)

$$+ \sum_{s=1}^{n}\left\{f^{(s+2)^*}\frac{1}{n}\left[N_{x,sub}^s(3)N_{x,tri}^{(s+1)^*}(3)\right]\right\} + \sum_{s=1}^{n}\left\{\frac{1}{n}N_{x,sub}^s(1)\sum_{t=1}^{n}f^t N_x^{i,t}\right\}.$$

First, we isolate the neighbour function $N_{xx}^i$ of vertex $i$ itself:

$$N_{xx}^i = \frac{1}{n}\sum_{s=1}^{n}\left\{N_{x,sub}^s(1)N_x^i + N_{x,sub}^s(2)N_{x,tri}^s(1) + N_{x,sub}^s(3)N_{x,tri}^{(s+1)^*}(1)\right\}.$$ (A17)

Then, we rearrange the sum terms containing $f^{(s+1)^*}$ by lowering the summing index by one:

$$\sum_{s=1}^{n} f^{(s+1)^*}\frac{1}{n}\left[N_{x,sub}^s(2)N_{x,tri}^s(3) + N_{x,sub}^s(3)N_{x,tri}^{(s+1)^*}(2)\right]$$

$$= \sum_{s=1}^{n} f^s\frac{1}{n}\left[N_{x,sub}^{(s-1)^*}(2)N_{x,tri}^{(s-1)^*}(3) + N_{x,sub}^{(s-1)^*}(3)N_{x,tri}^s(2)\right].$$ (A18)

We do the same for the term containing $f^{(s+2)^*}$:

$$\sum_{s=1}^{n} f^{(s+3)^*}\frac{1}{n}N_{x,sub}^s(3)N_{x,tri}^{(s+1)^*}(3) = \sum_{s=1}^{n} f^s\frac{1}{n}N_{x,sub}^{(s-2)^*}(3)N_{x,tri}^{(s-1)^*}(3).$$ (A19)

Then, by observing that the inner sum in the last term in Eq. A16 does not depend on the index of the outer sum, we arrange this term to:

$$\sum_{s=1}^{n}\left\{\frac{1}{n}N_{x,sub}^s(1)\sum_{t=1}^{n}f^t N_x^{i,t}\right\} = \sum_{s=1}^{n}\left[f^s N_x^{i,s}\right]\sum_{t=1}^{n}\left[\frac{1}{n}N_{x,sub}^t(1)\right].$$ (A20)

Substituting Eqs. A17 – A20 into Eq. A16 yields:

$$f_{xx}^i = f^i N_{xx}^i + \sum_{s=1}^{n} f^s\frac{1}{n}\left[N_{x,sub}^s(2)N_{x,tri}^s(2) + N_{x,sub}^{(s-1)^*}(2)N_{x,tri}^{(s-1)^*}(3) + N_{x,sub}^{(s-1)^*}(3)N_{x,tri}^s(2)\right.$$

$$\left. + N_{x,sub}^{(s-2)^*}(3)N_{x,tri}^{(s-1)^*}(3) + N_x^{i,s}\sum_{t=1}^{n}N_{x,sub}^t(1)\right]$$

$$= f^i N_{xx}^i + \sum_{s=1}^{n} f^s \frac{1}{n} \left[ N_{x,tri}^s(2) \left( N_{x,sub}^s(2) + N_{x,sub}^{(s-1)^*}(3) \right) \right.$$

$$\left. + N_{x,tri}^{(s-1)^*}(3) \left( N_{x,sub}^{(s-1)^*}(2) + N_{x,sub}^{(s-2)^*}(3) \right) + N_x^{i,s} \sum_{t=1}^{n} N_{x,sub}^t(1) \right]. \tag{A21}$$

As we see, this simplifies into the same form as Eq. A8:

$$f_{xx}^i = f^i N_{xx}^i + \sum_{s=1}^{n} f^s N_{xx}^{i,s}. \tag{A22}$$

The neighbour functions for the second derivative $f_{xx}^i$ (and, following the same derivation, $f_{xy}^i$ and $f_{yy}^i$) of $f$ on a non-boundary vertex $i$ are therefore given by:

$$N_{xx}^i = \frac{1}{n} \sum_{s=1}^{n} \left\{ N_{x,sub}^s(1) N_x^i + N_{x,sub}^s(2) N_{x,tri}^s(1) + N_{x,sub}^s(3) N_{x,tri}^{(s+1)^*}(1) \right\}, \tag{A23a}$$

$$N_{xy}^i = \frac{1}{n} \sum_{s=1}^{n} \left\{ N_{y,sub}^s(1) N_x^i + N_{y,sub}^s(2) N_{x,tri}^s(1) + N_{y,sub}^s(3) N_{x,tri}^{(s+1)^*}(1) \right\}, \tag{A23b}$$

$$N_{yy}^i = \frac{1}{n} \sum_{s=1}^{n} \left\{ N_{y,sub}^s(1) N_y^i + N_{y,sub}^s(2) N_{y,tri}^s(1) + N_{y,sub}^s(3) N_{y,tri}^{(s+1)^*}(1) \right\}, \tag{A23c}$$

$$N_{xx}^{i,s} = \frac{1}{n} \left[ N_{x,tri}^s(2) \left( N_{x,sub}^s(2) + N_{x,sub}^{(s-1)^*}(3) \right) + N_{x,tri}^{(s-1)^*}(3) \left( N_{x,sub}^{(s-1)^*}(2) + N_{x,sub}^{(s-2)^*}(3) \right) \right.$$

$$\left. + N_x^{i,s} \sum_{t=1}^{n} N_{x,sub}^t(1) \right], \tag{A23d}$$

$$N_{xy}^{i,s} = \frac{1}{n} \left[ N_{x,tri}^s(2) \left( N_{y,sub}^s(2) + N_{y,sub}^{(s-1)^*}(3) \right) + N_{x,tri}^{(s-1)^*}(3) \left( N_{y,sub}^{(s-1)^*}(2) + N_{y,sub}^{(s-2)^*}(3) \right) \right.$$

$$\left. + N_x^{i,s} \sum_{t=1}^{n} N_{y,sub}^t(1) \right], \tag{A23e}$$

$$N_{yy}^{i,s} = \frac{1}{n} \left[ N_{y,tri}^s(2) \left( N_{y,sub}^s(2) + N_{y,sub}^{(s-1)^*}(3) \right) + N_{y,tri}^{(s-1)^*}(3) \left( N_{y,sub}^{(s-1)^*}(2) + N_{y,sub}^{(s-2)^*}(3) \right) \right.$$

$$\left. + N_y^{i,s} \sum_{t=1}^{n} N_{y,sub}^t(1) \right]. \tag{A23f}$$

### A.3 First-order partial derivatives on staggered vertices

In order to describe fluxes between adjacent vertices, we need to define a discretisation of the derivatives on a point lying in between those two vertices. While it is possible to just take the average value of the derivatives on the vertices themselves, as

defined by Eq. A8, the resulting value would be a linear combination of the function values on all the neighbours of these two vertices, resulting in an undesirable degree of numerical diffusion.

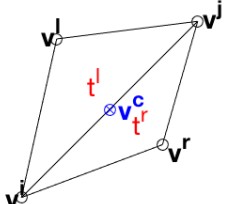

**Figure A3: a very simple mesh, showing the staggered vertex $v^c$ connecting vertices $v^i$ and $v^j$.**

5   Consider the simple mesh in Fig. A3, consisting of four vertices spanning two triangles. We defined the staggered vertex $v^c$ as the midpoint between vertices $v^i$ and $v^j$. On the two adjacent triangles $t^l$ and $t^r$, the derivatives $f_x^l, f_y^l, f_x^r$ and $f_y^r$ are well-defined:

$$f_x^l = \frac{-n_x^l}{n_z^l} = \frac{f^i(y^j - y^l) + f^j(y^l - y^i) + f^l(y^i - y^j)}{x^i(y^j - y^l) + x^j(y^l - y^i) + x^l(y^i - y^j)}, \tag{A26a}$$

$$f_y^l = \frac{-n_y^l}{n_z^l} = \frac{f^i(x^l - x^j) + f^j(x^i - x^l) + f^l(x^j - x^i)}{x^i(y^j - y^l) + x^j(y^l - y^i) + x^l(y^i - y^j)}, \tag{A26b}$$

$$f_x^r = \frac{-n_x^r}{n_z^l} = \frac{f^i(y^r - y^j) + f^j(y^i - y^r) + f^r(y^j - y^i)}{x^i(y^r - y^j) + x^j(y^i - y^r) + x^r(y^j - y^i)}, \tag{A26c}$$

$$f_x^r = \frac{-n_y^r}{n_z^l} = \frac{f^i(x^j - x^r) + f^j(x^r - x^i) + f^r(x^i - x^j)}{x^i(y^r - y^j) + x^j(y^i - y^r) + x^r(y^j - y^i)}. \tag{A26d}$$

Similar to the approach on the regular rectangular Arakawa C grid, we define the derivatives $f_x^c$ and $f_y^c$ of $f$ on $v^c$ as the average of the values on the two adjacent triangles:

$$f_x^c = \frac{1}{2}(f_x^l + f_x^r), \tag{A27a}$$

$$f_y^c = \frac{1}{2}(f_y^l + f_y^r). \tag{A27b}$$

10   The purpose of the Arakawa C grid is to separate velocities on cell boundaries into components parallel and orthogonal to those boundaries. The component orthogonal to the boundary carries a flux from one cell into another, while the parallel component does not. On the mesh, this is complicated by the fact that the orientation of boundaries is not fixed. Let **u** be the vector pointing from $v^i$ to $v^j$:

$$\boldsymbol{u} = \begin{bmatrix} x_j - x_i \\ y_j - y_i \end{bmatrix}. \tag{A28}$$

The parallel and orthogonal derivatives $f_{par}^c$ and $f_{ort}^c$ of $f$ on $v^c$ are then given by:

$$f_{par}^c = f_x^c \frac{u_x}{|u|} + f_y^c \frac{u_y}{|u|}, \tag{A29a}$$

$$f_{ort}^c = f_x^c \frac{-u_y}{|u|} + f_y^c \frac{u_x}{|u|}. \tag{A29b}$$

It can be shown that this results in:

$$f_{par}^c = \frac{f^j - f^i}{|u|}. \tag{A30}$$

This is, of course, simply the slope of the line between $v^i$ and $v^j$.

### A.4 Convergence of the discretisation scheme

The discretisation errors and convergence behaviour of the neighbour functions are shown in Figs. A4 and A5.

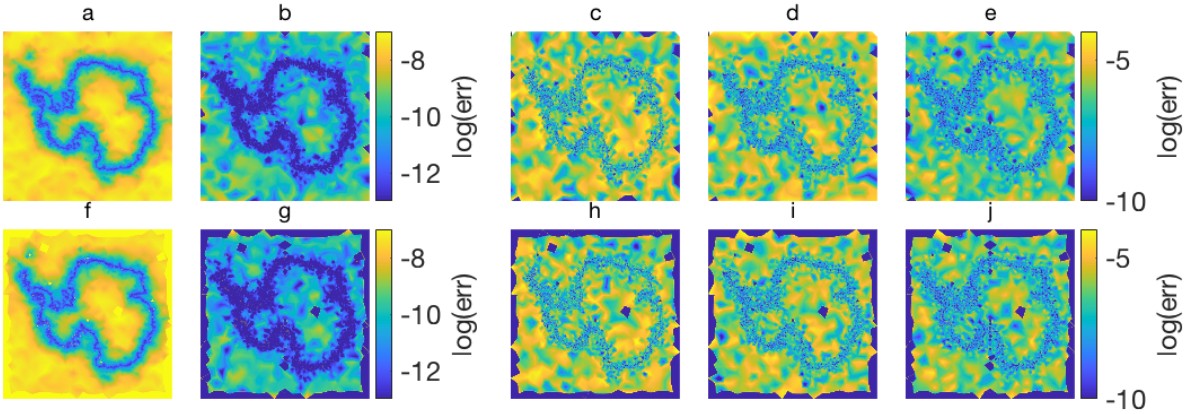

**Figure A4, panels a-e: discretisation errors for the averaged-gradients approach used in UFEMISM for, from left to right, $f_x$, $f_y$, $f_{xx}$, $f_{xy}$ and $f_{yy}$. Panels f-j: the same for the least-squares approach from Syrakos et al. (2017).**

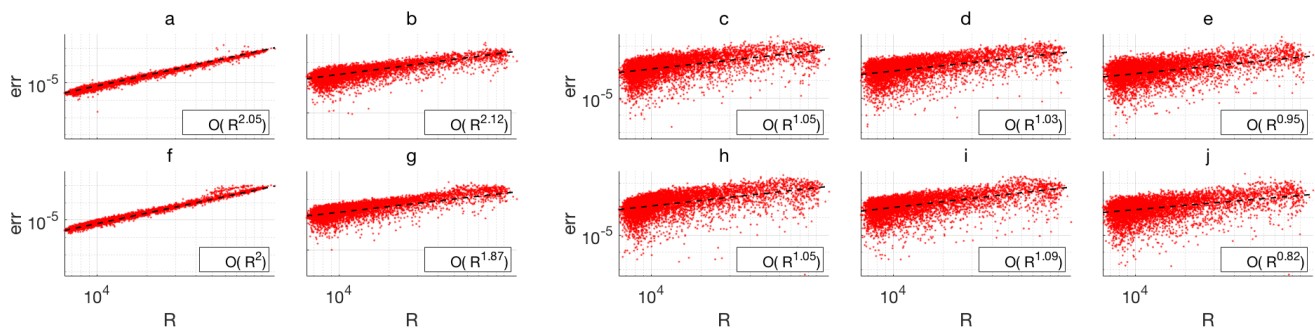

10    **Figure A5, panels a-e: discretisation errors vs. mesh resolution for the averaged-gradients approach used in UFEMISM for, from left to right, $f_x$, $f_y$, $f_{xx}$, $f_{xy}$ and $f_{yy}$. Panels f-j: the same for the least-squares approach from Syrakos et al. (2017). Loglinear fits have been made to all errors, with the order of convergence shown in the legends. Both approaches show very similar absolute errors and rates of convergence.**

Here, the derivatives of a simple function are calculated on a mesh generated by UFEMISM for Antarctica, and compared to

15    their analytical value. The absolute errors are shown in Fig. A4, clearly showing the dependence on grid resolution (which is

finest at the grounding line). This dependence is shown in Fig. A5, with loglinear curves fitted for all five derivatives showing the expected 2$^{nd}$- and 1$^{st}$-order convergence for the first and second derivatives, respectively. For comparison, the same errors and convergence rates are shown for the least-squares approach derived by Syrakos et al. (2017).

To demonstrate convergence in the ice-sheet model itself, Fig. A6 shows the error in the modelled steady-state ice margin radius for EISMINT experiment I, which decreases linearly with resolution. This matches the results of the previous section, as the SIA is a diffusive equation, meaning that the change in ice thickness is dictated strongly by the curvature of the surface (a second-order partial derivative, which the earlier convergence experiment shows to be first-order convergent).

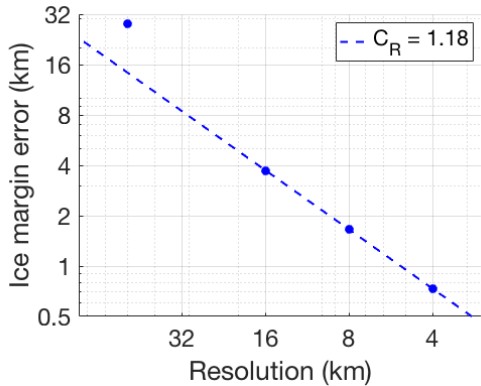

**Figure A6: modelled ice margin error vs. ice margin resolution for EISMINT experiment I. The dashed line is a log-linear fit of the form $ae^{C_R R}$, showing that the error decreases with resolution to approximately the first order.**

**Appendix B – finite volume approach to the conservation law**

In UFEMISM, ice thickness changes over time are calculated using a finite volume approach, which is conceptually very similar to the more commonly used combination of finite differencing with a staggered grid. Consider the conservation law

for flowing ice, equating the time derivative of the ice thickness $H$ to the (two-dimensional) divergence of the ice flux (being the product of $H$ and the vertically averaged ice velocity $\boldsymbol{v}$), and the balances $M$:

$$\frac{\partial H}{\partial t} = -\boldsymbol{\nabla} \cdot (\boldsymbol{v}H) + M. \tag{B1}$$

By applying the divergence theorem, this can be rewritten as:

$$\frac{\partial \overline{H_\Omega}}{\partial t} = \frac{-1}{A_\Omega} \oint_{\partial\Omega} (\boldsymbol{v}H \cdot d\widehat{\boldsymbol{n}}) + \overline{M}. \tag{B2}$$

Here, $\Omega$ is some arbitrary 2-D region (the control "volume" after which the finite volume approach is named), enclosed by the 1-D curve $\partial\Omega$ with outward unit normal vector $\widehat{\boldsymbol{n}}$. The unstructured triangular mesh partitions the 2-D domain into Voronoi

cells, which function as the control volumes (see Fig. B1).

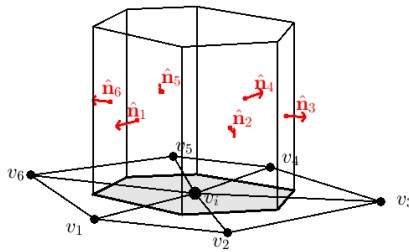

**Figure B1: the Voronoi cell (grey) of a vertex serves as the control volume in the finite volume approach. Ice flows through the vertical faces of the volume, while the surface and basal mass balance add or remove ice from the top and bottom faces, respectively.**

The conservation law for a single Voronoi cell reads:

$$\frac{\partial \overline{H_i}}{\partial t} = \frac{-1}{A_i} \sum_{c=1}^{n} \left[ \int_{\partial_c} (\boldsymbol{v_c} H_c \cdot d\widehat{\boldsymbol{n_c}}) \right] + \overline{M_i}. \tag{B3}$$

Here, the Voronoi cell of vertex $i$ shares the boundary $\partial_c$ with that of neighbouring vertex $c$. The equation is then discretised in space by assuming that the ice velocity $\boldsymbol{v}$ and the ice thickness $H$ are constant on $\partial_c$, so that the line integral becomes a simple multiplication with the length $L_c$ of the shared boundary $\partial_c$:

$$\frac{\partial H_i}{\partial t} = \frac{-1}{A_i} \sum_{c=1}^{n} [L_c \boldsymbol{v_c} H_c \cdot d\widehat{\boldsymbol{n_c}}] + \overline{M_i}. \tag{B4}$$

Lastly, the equation is discretised explicitly in time:

$$H_i^{t+\Delta t} = H_i^t + \Delta t \left[ \frac{-1}{A_i} \sum_{c=1}^{n} [L_c \boldsymbol{v_c} H_c \cdot d\widehat{\boldsymbol{n_c}}] + \overline{M_i} \right]. \tag{B5}$$

In order to solve this equation, we need to know the ice velocities $\boldsymbol{v_c}$ on the Voronoi cell boundaries. Both the SIA and the SSA are therefore solved on the staggered vertices described in Appendix A. The staggered ice thickness $H_c$ is determined using an up-wind scheme ($H_c = H_i$ if ice flows from $i$ to $j$, $H_c = H_j$ if it flows from $j$ to $i$). This means that the finite volume approach is essentially identical to the "mass-conserving up-wind finite difference scheme" used in PISM (Winkelmann et al., 2011), and very similar to the combination of finite differences with a staggered grid used in many other ice-sheet models.

The SIA and SSA are solved asynchronously, with the time steps determined as a function of the local resolution $R_c$ (defined as the distance between the two regular vertices connected by the staggered vertex $v_c$), the staggered SIA diffusivity $D_c$, and the staggered SSA ice velocities $u_c$, $v_c$, similar to the approach used in PISM (Bueler et al., 2007:

$$\Delta t_{SIA} = \min_c \left( \frac{-R_c^2}{6\pi D_c} \right), \tag{B6a}$$

$$\Delta t_{SSA} = \min_c \left( \frac{R_c}{|u_c| + |v_c|} \right), \tag{B6b}$$

**Appendix C – deriving a SOR iteration for solving the SSA**

A concrete form of the stress balance resulting from the SSA is given by Bueler and Brown (2009), based on the work by MacAyeal (1989) and Weis et al. (1999):

$$\frac{\partial}{\partial x}\left[2\bar{v}H\left(2u_x + v_y\right)\right] + \frac{\partial}{\partial y}\left[\bar{v}H\left(u_y + v_x\right)\right] - \frac{\tau_c u}{|\boldsymbol{u}|} = \rho g H h_x, \tag{C1a}$$

$$\frac{\partial}{\partial x}\left[\bar{v}H\left(u_y + v_x\right)\right] + \frac{\partial}{\partial y}\left[2\bar{v}H\left(u_x + 2v_y\right)\right] - \frac{\tau_c v}{|\boldsymbol{u}|} = \rho g H h_y. \tag{C1b}$$

The first two terms on the left-hand sides of these equations describe the membrane stresses, in terms of the horizontal ice velocities $u$ and $v$, the vertically averaged ice viscosity $\bar{v}$ and the ice thickness $H$. The last terms on the left-hand side describe the basal stress resulting from a Coulomb sliding law, in terms of the till yield stress $\tau_c$ (which is spatially variable, and is zero for floating ice) and the horizontal ice velocities $u$ and $v$. These stresses are balanced out by the driving stress, which is described on the right-hand sides in terms of the ice thickness $H$ and the surface slopes $h_x$ and $h_y$. In order to simplify Eq. C1 we follow the approach by Determann (1991) and Huybrechts (1992), where the lateral variations of the effective strain rate $\frac{\partial \eta}{\partial x}, \frac{\partial \eta}{\partial y}$ are neglected. Determann (1991) and Huybrechts (1992) show that, since these terms are small compared to variations of the individual strain rates, this does not significantly affect the solution, while improving numerical stability and computational efficiency. Using this simplification, and setting the basal shear stress $\tau_b = \frac{\tau_c}{|\boldsymbol{u}|}$, Eq. C1 simplifies to:

$$4u_{xx} + u_{yy} + 3v_{xy} - \frac{\tau_b u}{\bar{v}H} = \frac{\rho g h_x}{\bar{v}}, \tag{C3a}$$

$$4v_{yy} + v_{xx} + 3u_{xy} - \frac{\tau_b v}{\bar{v}H} = \frac{\rho g h_y}{\bar{v}}. \tag{C3b}$$

Although both $\bar{v}$ and $\tau_b$ are functions of $u$ and $v$, this dependence is relatively weak. Bueler and Brown (2009) show that a good approach to solving these equations, is to use two nested iterative loops: an inner loop that solves $u$ and $v$ for a given $\bar{v}$ and $\tau_b$, and an outer loop that updates $\bar{v}$ and $\tau_b$ with the new values of $u$ and $v$. They show that all four variables converge to a unique, stable solution. Since $\bar{v}$ and $\tau_b$ are not updated in the inner loop, the derivation for the relaxation iteration is greatly simplified. Replacing the derivatives of $u$ and $v$ in Eq. 3 with their discretised approximations (as derived in Appendix A) yields:

$$4\left(u^i N_{xx}^i + \sum_{c=1}^{n} u^c N_{xx}^c\right) + u^i N_{yy}^i + \sum_{c=1}^{n} u^c N_{yy}^c + 3v_{xy} - \frac{\tau_b u^i}{\bar{v}H} = \frac{\rho g h_x}{\bar{v}}, \tag{C4a}$$

$$4\left(v^i N_{yy}^i + \sum_{c=1}^{n} v^c N_{yy}^c\right) + v^i N_{xx}^i + \sum_{c=1}^{n} v^c N_{xx}^c + 3u_{xy} - \frac{\tau_b v^i}{\bar{v}H} = \frac{\rho g h_y}{\bar{v}}. \tag{C4b}$$

Isolating $u^i$ and $v^i$ yields:

$$u^i\left(4N_{xx}^i + N_{yy}^i - \frac{\tau_b}{\bar{\nu}H}\right) + \sum_{c=1}^{n} u^c\left(4N_{xx}^c + N_{yy}^c\right) + 3v_{xy} = \frac{\rho g h_x}{\bar{\nu}}, \tag{C5a}$$

$$v^i\left(4N_{yy}^i + N_{xx}^i - \frac{\tau_b}{\bar{\nu}H}\right) + \sum_{c=1}^{n} v^c\left(4N_{yy}^c + N_{xx}^c\right) + 3u_{xy} = \frac{\rho g h_y}{\bar{\nu}}. \tag{C5b}$$

Defining the "centre coefficients" $e_u^i = 4N_{xx}^i + N_{yy}^i - \frac{\tau_b}{\bar{\nu}H}$ and $e_v^i = 4N_{yy}^i + N_{xx}^i - \frac{\tau_b}{\bar{\nu}H}$, and rearranging the terms in Eq. 5 yields:

$$u^i = \frac{1}{e_u^i}\left[\frac{\rho g h_x}{\bar{\nu}} - 3v_{xy} - \sum_{c=1}^{n} u^c\left(4N_{xx}^c + N_{yy}^c\right)\right], \tag{C6a}$$

$$v^i = \frac{1}{e_v^i}\left[\frac{\rho g h_y}{\bar{\nu}} - 3u_{xy} - \sum_{c=1}^{n} v^c\left(4N_{yy}^c + N_{xx}^c\right)\right]. \tag{C6b}$$

These equations now express the function values $u^i$ and $v^i$ in terms of the values of their direct neighbours, in the form $u^i = f(u^{n1}, u^{n2}, \dots, u^n)$ this can be adapted into an SOR iteration of the form $u_i^{t+1} = (1 - \omega)u_i^t + \omega f(u^{n1}, u^{n2}, \dots, u^n)$. This

5   yields the following equations:

$$u_i^{t+1} = (1 - \omega)u_i^t + \frac{\omega}{e_u^i}\left[\frac{\rho g h_x}{\bar{\nu}} - 3v_{xy} - \sum_{c=1}^{n} u^c\left(4N_{xx}^c + N_{yy}^c\right)\right], \tag{C7a}$$

$$v_i^{t+1} = (1 - \omega)v_i^t + \frac{\omega}{e_v^i}\left[\frac{\rho g h_y}{\bar{\nu}} - 3u_{xy} - \sum_{c=1}^{n} v^c\left(4N_{yy}^c + N_{xx}^c\right)\right]. \tag{C7b}$$

Rearranging these terms gives the equations that can be found in the UFEMISM source code:

$$u_i^{t+1} = u_i^t - \frac{\omega}{e_u^i}\left[\sum_{c=1}^{n} u^c\left(4N_{xx}^c + N_{yy}^c\right) + e_u^i u_i^t + 3v_{xy} - \frac{\rho g h_x}{\bar{\nu}}\right], \tag{C8a}$$

$$v_i^{t+1} = v_i^t - \frac{\omega}{e_v^i}\left[\sum_{c=1}^{n} v^c\left(4N_{yy}^c + N_{xx}^c\right) + e_v^i v_i^t + 3u_{xy} - \frac{\rho g h_y}{\bar{\nu}}\right]. \tag{C8b}$$

Finding the optimal value for the relaxation parameter $\omega$, to achieve the highest possible convergence without creating numerical instability, is one of the most-studied subjects in numerical mathematics. In the experiments described in Sects. 3 and 4, a value of 1.1 was used; in the schematic MISMIP experiments higher values were often possible, resulting in

10   substantially faster convergence, but this sometimes resulted in instability in the realistic experiments, which have steeper gradients in surface slopes and ice viscosity. Preliminary experiments with UFEMISM have suggested that, when $\omega$ is too high, numerical instability primarily occurs at vertices that are surrounded by sharp triangles. This suggests that, in order to optimise computational performance, $\omega$ could be made into a spatially variable field $\omega_i$, with values depending on mesh geometry. Further preliminary experiments showed that, while this does indeed significantly improve performance, it is

difficult to create a robust setup that is guaranteed to either always avoid, or otherwise cope with, numerical instability. Solving this problem is deemed to be beyond the scope of the current study.

Since ice velocities are desired on the staggered vertices, this is where the SSA must be solved. However, the neighbour functions for staggered vertices, derived in Appendix A, only work for the first-order partial derivatives. With only four surrounding regular vertices, it is not possible to uniquely define all five first- and second-order partial derivatives. This problem is solved by solving the SSA on both the regular and the staggered vertices simultaneously (see Fig. C1). Each staggered vertex is then surrounded by 6 neighbours (2 regular vertices and 4 staggered vertices), and both the first and second derivatives can be discretised using neighbour functions. The SSA, which includes the second derivatives of the ice velocities, can then be solved on the staggered vertices, so that it can be used in the finite volume approach.

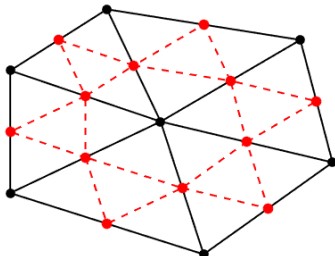

**Figure C1: by combining the regular (black) and staggered (red) vertices into a single mesh, each staggered vertex has 6 direct neighbours, so that neighbour functions for both the first and second derivatives can be calculated, using the approach from Appendix A.**

The semi-analytical solution to the grounding-line flux by Tsai et al. (2015) given by Eq. 4 is implemented as a boundary condition to achieve better grounding-line dynamics. This is done by designating staggered vertices as grounding-line vertices if they lie between a grounded and a floating regular vertex. If this is the case, the analytical ice flux solution is calculated for this staggered grounding-line vertex (finding the sub-grid grounding-line ice thickness by interpolating the thickness above flotation between the two regular vertices). The flux is divided by the ice thickness to find the velocity, which is assumed to have a direction antiparallel to the gradient of the thickness above flotation (i.e. perpendicular to the grounding line). During the SOR iteration, ice velocities at these staggered grounding-line vertices are simply kept fixed to this analytical solution. Other implementations that do not alter the SSA solution, but rather overwrite the ice flux in the free surface update, require a "heuristic" to determine which grid cell to apply the flux to (an approach which violates conservation of mass). Our approach circumvents this problem.

**Appendix D – deriving the iterative scheme for solving the heat equation**

In order to calculate a depth-dependent temperature distribution, a vertical spatial discretisation is required. Following the approach used by many ice-sheet models, we adopt the scaled vertical coordinate:

$$\zeta = \frac{h - z}{H}. \tag{D1}$$

This guarantees that the top and bottom of the vertical ice column always coincide with the first and last vertical grid point, respectively. This coordinate transformation results in the appearance of a few extra terms in the heat equation (Eq. 5):

$$\frac{\partial T}{\partial t} + \frac{\partial T}{\partial \zeta}\frac{\partial \zeta}{\partial t} = \frac{k}{\rho c_p}\frac{\partial^2 T}{\partial \zeta^2}\left(\frac{\partial \zeta}{\partial z}\right)^2 - u\left(\frac{\partial T}{\partial x} + \frac{\partial T}{\partial \zeta}\frac{\partial \zeta}{\partial x}\right) - v\left(\frac{\partial T}{\partial y} + \frac{\partial T}{\partial \zeta}\frac{\partial \zeta}{\partial y}\right) - w\left(\frac{\partial T}{\partial \zeta}\frac{\partial \zeta}{\partial z}\right) + \frac{\Phi}{\rho c_p}. \tag{D2}$$

The different spatial derivatives of $\zeta$ follow from Eq. C1:

$$\frac{\partial \zeta}{\partial t} = \frac{1}{H}\left(\frac{\partial h}{\partial t} - \zeta\frac{\partial H}{\partial t}\right), \tag{D3a}$$

$$\frac{\partial \zeta}{\partial x} = \frac{1}{H}\left(\frac{\partial h}{\partial x} - \zeta\frac{\partial H}{\partial x}\right), \tag{D3b}$$

$$\frac{\partial \zeta}{\partial y} = \frac{1}{H}\left(\frac{\partial h}{\partial y} - \zeta\frac{\partial H}{\partial y}\right), \tag{D3c}$$

$$\frac{\partial \zeta}{\partial z} = \frac{-1}{H}. \tag{D3d}$$

The scaled vertical derivatives of a function, e.g. $\frac{\partial T}{\partial \zeta}$ and $\frac{\partial^2 T}{\partial \zeta^2}$, can be discretised to yield expressions of the form:

$$\frac{\partial T_k}{\partial \zeta} = a_\zeta T_{k-1} + b_\zeta T_k + c_\zeta T_{k+1}, \tag{D4a}$$

$$\frac{\partial^2 T_k}{\partial \zeta^2} = a_{\zeta\zeta} T_{k-1} + b_{\zeta\zeta} T_k + c_{\zeta\zeta} T_{k+1}. \tag{D4b}$$

5   These expressions are very similar to the neighbour functions that were used in Appendix A. We will not include their derivation here. Using these expressions to discretise the vertical derivatives in Eq. D2, and using an implicit time-discretisation, yields:

$$\frac{T_k^{t+\Delta t} - T_k^t}{\Delta t} + \frac{\partial \zeta}{\partial t}\left(a_\zeta T_{k-1}^{t+\Delta t} + b_\zeta T_k^{t+\Delta t} + c_\zeta T_{k+1}^{t+\Delta t}\right) =$$

$$\frac{k}{\rho c_p H^2}\left(a_{\zeta\zeta} T_{k-1}^{t+\Delta t} + b_{\zeta\zeta} T_k^{t+\Delta t} + c_{\zeta\zeta} T_{k+1}^{t+\Delta t}\right)$$

$$-u\left[\frac{\partial \zeta}{\partial x}\left(a_\zeta T_{k-1}^{t+\Delta t} + b_\zeta T_k^{t+\Delta t} + c_\zeta T_{k+1}^{t+\Delta t}\right) + \frac{\partial T}{\partial x}\right]$$

$$-v\left[\frac{\partial \zeta}{\partial y}\left(a_\zeta T_{k-1}^{t+\Delta t} + b_\zeta T_k^{t+\Delta t} + c_\zeta T_{k+1}^{t+\Delta t}\right) + \frac{\partial T}{\partial y}\right] \tag{D5}$$

$$-w\left[\frac{-1}{H}\left(a_\zeta T_{k-1}^{t+\Delta t} + b_\zeta T_k^{t+\Delta t} + c_\zeta T_{k+1}^{t+\Delta t}\right)\right] + \frac{\Phi}{\rho c_p}.$$

This can be rearranged to read:

$$T_{k-1}^{t+\Delta t}\left[a_\zeta\left(\frac{\partial \zeta}{\partial t} + u\frac{\partial \zeta}{\partial x} + v\frac{\partial \zeta}{\partial y} - \frac{w}{H}\right) - \frac{a_{\zeta\zeta}k}{\rho c_p H^2}\right]$$

$$+T_k^{t+\Delta t}\left[b_\zeta\left(\frac{\partial\zeta}{\partial t}+u\frac{\partial\zeta}{\partial x}+v\frac{\partial\zeta}{\partial y}-\frac{w}{H}\right)-\frac{b_{\zeta\zeta}k}{\rho c_p H^2}-\frac{1}{\Delta t}\right]$$

$$+T_{k+1}^{t+\Delta t}\left[c_\zeta\left(\frac{\partial\zeta}{\partial t}+u\frac{\partial\zeta}{\partial x}+v\frac{\partial\zeta}{\partial y}-\frac{w}{H}\right)-\frac{c_{\zeta\zeta}k}{\rho c_p H^2}\right] \tag{D6}$$

$$=\frac{T_k^t}{\Delta t}-u\frac{\partial T}{\partial x}-v\frac{\partial T}{\partial y}+\frac{\Phi}{\rho c_p}.$$

This system of equations can be represented by the matrix equation $AT^{t+\Delta t}=\delta$, where the lower diagonal $\alpha$, central diagonal $\beta$ and upper diagonal $\gamma$ of the tridiagonal matrix $A$, and the solution $\delta$, are given by:

$$\alpha=a_\zeta\left(\frac{\partial\zeta}{\partial t}+u\frac{\partial\zeta}{\partial x}+v\frac{\partial\zeta}{\partial y}-\frac{w}{H}\right)-\frac{a_{\zeta\zeta}k}{\rho c_p H^2}, \tag{D7a}$$

$$\beta=b_\zeta\left(\frac{\partial\zeta}{\partial t}+u\frac{\partial\zeta}{\partial x}+v\frac{\partial\zeta}{\partial y}-\frac{w}{H}\right)-\frac{b_{\zeta\zeta}k}{\rho c_p H^2}-\frac{1}{\Delta t}, \tag{D7b}$$

$$\gamma=c_\zeta\left(\frac{\partial\zeta}{\partial t}+u\frac{\partial\zeta}{\partial x}+v\frac{\partial\zeta}{\partial y}-\frac{w}{H}\right)-\frac{c_{\zeta\zeta}k}{\rho c_p H^2}, \tag{D7c}$$

$$\delta=\frac{T_k^t}{\Delta t}-u\frac{\partial T}{\partial x}-v\frac{\partial T}{\partial y}+\frac{\Phi}{\rho c_p}. \tag{D7d}$$

This matrix equation has to be solved for every individual grid cell. In UFEMISM, this is done with the Fortran package LAPACK; in Matlab, it can be done with the "backslash method": `T = A\delta`. We apply a Dirichlet boundary condition at the top of the column, keeping ice temperature equal to surface air temperature. At the base, a Neumann boundary condition is applied, keeping the vertical temperature gradient fixed to a value dictated by the geothermal heat flux and the frictional heating from sliding. Ice temperature throughout the vertical column is limited to the depth-dependent pressure melting point.

**Appendix E – mesh generation**

UFEMISM uses a modified version of Ruppert's algorithm to generate the unstructured triangular meshes on which the model equations are solved. Ruppert's Algorithm (Ruppert, 1995) iteratively adds vertices to an existing Delaunay triangulation (a process called refinement), until the smallest internal angle of all triangles no longer lies below a pre-defined threshold value (typically 25 degrees). Since the stability of many numerical iterative schemes for solving differential equations strongly depends on this particular property of the mesh geometry (Ruppert, 1995), Ruppert's Algorithm is a very useful tool for generating meshes suitable for use in physical models. During mesh refinement, a triangle is marked as "bad" if its smallest internal angle lies below the threshold value. If this is the case, the triangle is "split", by adding a vertex at the triangle's circumcentre, and updating the Delaunay triangulation. Ruppert (1995) showed that, as long as the perimeter of the mesh does not contain any angles sharper than the threshold value, this algorithm always converges, i.e. results in a mesh with no "bad" triangles and a finite, typically small number of vertices.

In UFEMISM, several additional conditions have been added which can cause a triangle to be marked as "bad", which depend upon the triangle's "model content" and size. If, for example, the modelled grounding line passes through a triangle whose longest edge exceeds the specified maximum grounding line resolution, that triangle is also marked as "bad".

Fig. E1a shows the results of a simple schematic experiment, where an ice sheet and shelf are simulated on a semi-circular island, which has a small embayment on the south-east side (so that a small ice shelf forms). This configuration serves only to provide a clear separation of the different areas of interest, so that the behaviour of the mesh generation code can be investigated when different resolutions are prescribed for the grounding line, the calving front, the ice margin on land, and the coastline. Fig. E1b-f show the resolution distribution for vertices lying on these lines, with the prescribed resolutions shown by the

vertical dashed red lines. As can be seen, the resulting resolutions are very close to their desired values.

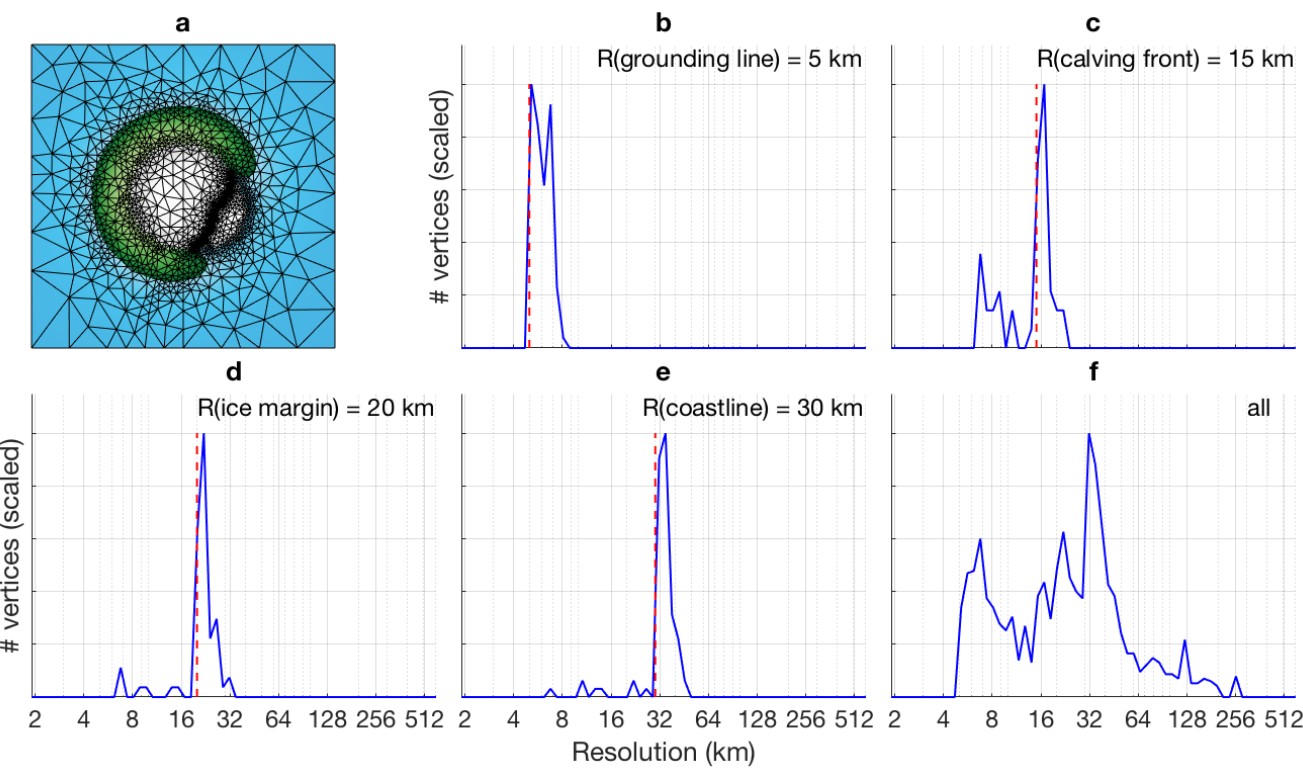

**Figure E1a: the results of a schematic experiment, where an ice sheet and shelf are simulated over a small island with an embayment on the southeast side. Blue indicates open ocean, green indicates ice-free land, white indicates ice. b-f: resolution distributions for, respectively, the grounding line, the calving front, the ice margin, the coastline, and the entire mesh. The resolutions that have been**
**prescribed for these different regions are indicated by the vertical dashed red lines.**

Extensive preliminary experiments showed that this particular approach to mesh generation is robust, resulting in a mesh with resolutions over the specified areas that lie very close to the specified value. Other conditions for mesh refinement that can be included, but will not be discussed here, include a maximum vertical error in surface elevation based on surface curvature (useful for the interior close to the ice margin), and a maximum resolution derived from ice velocity (useful for ice streams).

Lastly, a maximum resolution can be prescribed at specific geographic locations, such as ice core drill sites, which is demonstrated in Fig. 1. Since future plans for UFEMISM include the addition of a tracer tracking module, this feature will be very useful for creating pseudo-ice cores which can be directly compared to observations. The resolutions shown in Fig. E1 were chosen simply as a demonstration. As far as we've been able to find in preliminary experiments, there is no hard limit to

how fine a resolution can be prescribed. In practise, the resolution for palaeo-simulations will always be limited by the computation time

Mesh generation has been parallelised by subdividing the model domain into separate regions for each processor, which generates a mesh for only that section. These "submeshes" are then merged to produce one single mesh that covers the entire

model domain. An example of four such submeshes that have been generated in parallel for Antarctica is shown in Fig. E2. Although these particular submeshes have equally-sized domains (for illustration purposes), in general the size of each subdomain is chosen such that they contain approximately equal numbers of vertices, to ensure proper load balancing. The results presented in Sect. 4 show that this approach results in a computation time that scales with the number of processors to order 0.47. Note that this form of domain partitioning is only used in the parallelisation of the mesh generation. Once a mesh

has been generated, the use of MPI shared memory means that every processor can access all data on the entire mesh, and operations are partitioned simply by number of vertices.

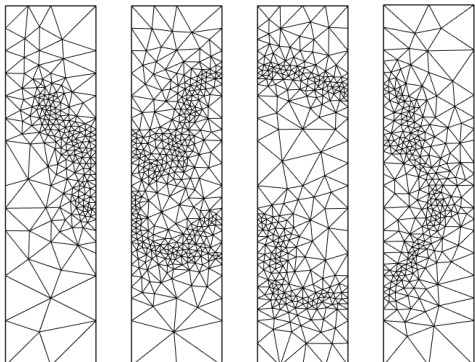

**Figure E2: four Antarctic submeshes that will be merged into a single, domain-wide mesh**

The mesh adapts to the ice-sheet geometry as it evolves through time. During a simulation, a "mesh fitness factor", defined as

the fraction of triangles that still meet all the fitness criteria in the extended version of Ruppert's algorithm, is calculated periodically (with a time step that can be specified through the config file, typically 50 years). For a newly generated mesh, this fraction is, by definition, 100 %. As the ice-sheet geometry evolves over time, the fitness factor slowly decreases. When it falls below a prescribed threshold value (typically 95 %), a mesh update is triggered, and an entire new mesh is generated. In glacial cycle simulations, the typical time between mesh updates is 50 – 500 model years. Although Ruppert's algorithm

provides a very intuitive rule for refining a mesh, no such rules can be easily defined for "un-refining" a mesh. The simplest approach is therefore to generate an entirely new mesh once the fitness of the current mesh falls below the threshold value.

Fig. E3 shows the meshes of the Antarctic retreat simulation shown in Fig. 11, zoomed in on the Ross ice shelf. Here we see how the high-resolution sections of the mesh follow the retreating ice sheet, ensuring the grounding line is always resolved at the desired resolution. The parallelisation by domain subdivision is also visible; the borders between the domains of the different processors are indicated by the red lines.

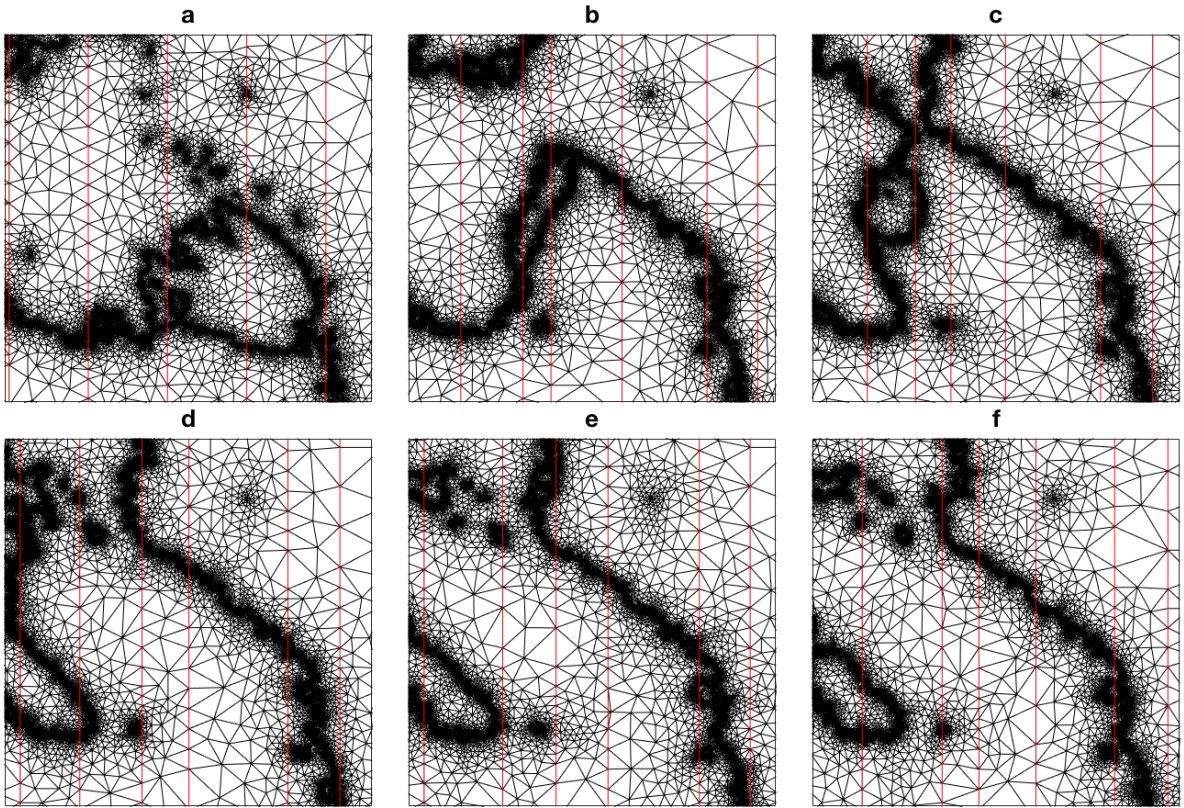

**Fig. E3: the meshes of the Antarctic retreat simulation shown in Fig. 11, zoomed in on the Ross ice shelf. The retreat of the ice sheet is clearly visible in the mesh structure, as are the processor domains resulting from the load-balanced domain subdivision (borders highlighted in red).**

Once a new mesh has been generated, the ice thickness and englacial temperature are remapped from the old mesh to the new one, using conservative remapping based on the method by Jones (1999), adapted from spherical to Cartesian coordinates. Extensive preliminary experiments have shown that conservative remapping is crucial for achieving accurate results; using simple trilinear interpolation, or other easily implemented approaches, results in ice sheets that deviate significantly from the analytical solutions of the Halfar and Bueler benchmark experiments discussed in Sect. 3.1, especially when a high resolution is used (since this results in more frequent mesh updates). In the EISMINT experiments, ice thickness is less affected since this is mostly dictated by the mass balance forcing, but large errors occur in the englacial temperature when non-conservative remapping schemes are used.

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
