# Peer review of "The Utrecht Finite Volume Ice-Sheet Model: UFEMISM (version 1.0)"

_Geoscientific Model Development, 2020_

## Referee Comment (RC1) · Josefin Ahlkrona (Referee) · 14 Oct 2020

General comments
* * *
The paper presents a new ice sheet model. The model equations are the shallow ice (SIA) and shallow shelf equations (SSA), coupled together using the method of Beuler and Brown (2009). The equations are discretised using a Finite Volume scheme derived by the authors, on an adaptive mesh. The paper is well written and the topic is well suited for GMD. The novel contribution is the use of the Finite Volume Method and to some extent the usage of mesh adaptivity in combination with a SIA/SSA model (although that has been implemented previously). The mesh adaptivity enables a high

resolution at the grounding line, but it is not clear to me that a high resolution is worth it as the model equations used are inaccurate at the grounding line. Nevertheless the study could be a interesting contribution if this issue is investigated deeper and if the accuracy and efficiency of the numerical scheme is evaluated more thoroughly. I appreciate that several verification experiments have been made, however, only part of the model is verified in experiments, convergence experiments testing the Finite Volume scheme are missing and more experiments are needed regarding the adaptive meshes. In particular since the numerical scheme is constructed by the authors, it is important to motivate the choice of this scheme and verify it thoroughly. The paper would also benefit from a clearer description of the model and the numerical scheme used. I recommend major revisions.

Specific comments
* * *
The SIA-SSA model is known to be inaccurate at the grounding line. A fine grid at the grounding line thus means that the numerical error is low, while the model error is inevitably high. Does it make sense to resolve the grounding line despite that the model errors remain, and if so, how much? Include experiments that shows the balance between model and numerical errors, by using a find grid full Stokes model as a reference solution. These experiments would be of value to all models using SIA-SSA at the grounding line.

The model is only validated for the SIA equations, not the SSA equations. In order for the community to trust the accuracy of the model, the full SIA/SSA model must also be evaluated in experiments in some way. As the authors state, this is more difficult due to the lack of an analytical solution. However, the numerical discretisation can be tested by running convergence experiments, using a fine mesh solution as a reference. Does the errors decrease as expected when the mesh is refined?

The Finite Volume Method is an unusual choice for an ice sheet model, and a motivation of this choice should be given. Add a summary of the method in the main paper, including a statement of the flux approximation (the specific details can remain in the appendix) and the order of the scheme. Discuss if numerical mass conservation is important compared to the model errors, preferably illustrated with experiments. As already mentioned in point 2), include convergence experiments. The convergence experiments in Figure 3-10 only show THAT the model converges - not if it converges with at the expected rate.

Include experiments showing how the mesh adaptivity works. The results shown in Figure 3-10 are done on a uniform grid. These should be done using mesh adaptivity, given that the mesh adaptivity is one of the main contributions of the paper. Experiment with choice of conditions for refinement in these experiments.

Page 2, Line 12: Include a reference to https://tc.copernicus.org/articles/10/307/2016/

Page 2, Line 15: Add a sentence explaining why fine resolution is more important when buttressing is significant.

Page 2, Line 20: Discuss how the time step relates to the mesh size in a non-uniform mesh.

Page 2, Line 28: Define what properly capturing grounding line dynamics means to you, given that the model equations you use do not include all stress components.

Page 3, Line 4: Optimal in which sense? I believe it is a compromise rather than an optimum.

Page 3, Line 5: Why is the model abbreviated UFEMISM, instead of UFVMISM? The name suggests that it is a finite element model.

Page 3, Section 2.1: State here that you use Fortran 90 and motivate this choice

Page 3, Line 20-21: Is there a limit to how fine mesh the user can choose? If so, why?

Page 3, Line 24: Elaborate on why this is important. Why is mass conservation important when the overall mass balance will never be accurate due to model errors and mesh resolution limits?

Page 3, Line 26: Is the number of vertical layers hard coded to 15?

Page 3, Line 30: There are other models that solve simplified equations suited for paleo models and also use mesh adaptivity, for instance the SIA/SSA mode of Elmer/Ice. Include references to these models.

Page 4, Line 12: Specify what is meant by "model content"

Page 4. Line 7-23: Do include a discussion the impact of different choices of conditions for refining the mesh, how do they impact the accuracy of different variables? These kind of conditions makes the mesh refinement interesting.

Page 5, Figure 1: The mesh extends into the ocean. Are there passive volumes out side the ice domain? Clarify.

Page 5, Line 16: Is the partitioning updated as the mesh is updated to maintain proper load balancing? If so, how? If not, motivate why this is not needed.

Page 6, Line 10. Did you consider simply deforming the mesh instead of generating a new mesh, or using deformation to decrease the frequency with which the mesh has to be updated.

Page 7, Line 1: The SIA is formulas to evaluate rather than equations to solve, since the velocity and pressure is already solved for.

Page 7, Line 2-4: It could be potentially confusing to discuss Finite Difference schemes while the model in the paper is based on Finite Volumes. I suggest clarifying this paragraph by beginning the discussion with a couple of sentences explaining the conceptual difference between FD and FVM, then move on to describing how the flux is approximated, and lastly relate this to the FD references and Type I / Type II models.

Page 7, Line 2-4: Add some references to FVM literature and relate your flux approximation to existing schemes.

Page 7, Line 2-4: I would like to see the formulas of ice fluxes and how they are used to calculated SIA velocities here, since the FVM is usually used for solving equations rather than calculating formulas.

Page 7: Write down not only the SIA formulas but also the SSA equations for consistency, together with a brief explanation of how they are merged.

Page 7, Line 9: Add a summary of the FVM approach for solving the SSA.

Page 7: State the theoretical order of accuracy of the FVM scheme

Page 7, Line 13: Clarify that it is Finite Differences in time (not space)

Page 7: Add reference to strain heating form (equation 4)

Page 7, Line 20: Is 15 layers hard coded?

Page 7, section 2.4: What is the spatial discretisation of the temperature equation?

Section 2: Specify how the free surface variable h is updated

Section 2: Specify which equation is limiting the time step - the free surface or the temperature equation?

Section 3: Elaborate on the purpose of each experiments Experiments section 3: These should be done using adaptive meshes.

Page 9, Line 6-7: This sentence needs clarification. As you are looking at numerical error, clarify how the surface slope impacts it so that it is clearer to the reader what is model error and numerical error

Page 9, Line 10: Does it make sense to reduce the error to an error of 10 km, when the SIA is very inaccurate at the margin? How large is the model error?

Figure 5-8: Can you include results from the EISMINT benchmark for in the figure for

Interactive
comment

comparison?

Section 3.2: Explain the EISMINT experiment in such a way that a reader does not have to be familiar with the original publication. Describe e.g. how a fixed versus moving margin is achieved.

Results in Figure 9 and 10: Would it be possible to instead look at the margin, since this is where the resolution matters?

Page 15, Figure 12: Why is SIA so expensive?

Page 15, Figure 12: In which part is the free surface update included?

Page 16, paragraph 1: Discuss why you think mesh update scales less well

Page 16, Line 7: Move the definition of b_r to the next paragraph

Page 17, Line 8: A dynamic time step is mentioned here. An ellaboration on the dynamic time step for SIA/SSA should be added in the model description

Page 17, Line 11-26: Do this experiment with some of the conditions for refinement that were discussed on page 4 rather than explicitly setting the refinement

Page 17, Line 11-26: These results need to be presented more in detail in a figure or table, since this experiment leads to the result that UFEMISM is 10-30 times faster than ANICE.

Page 18, line 5: The claim that the numerical scheme is valid should be supported by convergence experiments and tests of the SSA discretisation

Page 18, Line 11-26: Try to use a fine resolution full Stokes MISMIP solution as a reference solution, in order to evaluate the SIA/SSA model and the convergence of the numerical scheme

Appendix A: Why did you choose to go through this derivation instead of employing some standard FVM scheme? Why did you choose to start from a finite difference

scheme?

Appendix A: As FVM is not commonly used in ice sheet modelling, write down the basics of finite volumes and describe your scheme in that framework.

Technical corrections
* * *
Page 3, Line 20: "Paradox" -> compromise.

Page 3, Line 21: "config file" -> "configuration file"

Page 4, Line 2: " refinement " -> refinement

Page 4, Line 17-20: The sentence starting with "Other conditions" is a bit unclear and long

Page 10, Equation 10: Add spaces between formulas

Page 11, Table 1: Order the rows so that they are consistent with the order of the figures, or vice versa

Page 12, Line 14: Vertical discretisation -> Vertical resolution

Page 14, Line 11: "shared memory" -> shared memory

Figure 12: Include in the caption an explanation of what bc and br are.

---

## Referee Comment (RC2) · Ralf Greve (Referee) · 23 Dec 2020

In this manuscript, the authors introduce a new ice-sheet model called UFEMISM. It is based on hybrid SIA-SSA dynamics for grounded ice, SSA dynamics for floating ice, and employs finite-volume techniques on an adaptive mesh for solving the model equations numerically. The SIA part of the model is verified against analytical solutions and previous results from EISMINT. An application to the Antarctic ice sheet illustrates the capability of the model to deal with real-world problems and the computing time demands depending on the number of cores and the resolution.

While the new model is very interesting, I find the paper pretty much incomplete. The most severe omission is that no attempt is made to verify the SSA part of the

model, and the performance of the full model with respect to grounding line dynamics. Granted, analytical solutions are lacking. However, model intercomparisons have been done within the several MISMIP initiatives, most recently MISMIP+ by Cornford et al. (2020, doi: 10.5194/tc-14-2283-2020). To enable the reader to appreciate the performance of the new model, it would really be crucial to carry out such types of experiments and demonstrate how the model behaves in terms of grounding line advance and retreat as a function of resolution.

In Section 2.3, the short description of the ice-thickness solver only applies to the SIA, for which the ice-thickness equation can be written in diffusion form. How is the general ice-thickness equation (that includes the SSA part of the dynamics) solved?

In Section 4, the description of the experiments, given in a mere four lines (p. 14, l. 15-18), lacks detail. What are the initial conditions for the experiments? What are the physical parameters (rate factor, basal sliding law, heat conductivity & capacity, geothermal heat flux, etc.)? What is assumed for ice-shelf basal melting? The tested resolutions should be mentioned in the text, not only in the figure captions. What happens to the ice sheet in a control scenario (no warming applied)? I am actually quite surprised that the ice sheet reacts so strongly on just a surface warming without (I presume) changes in the SMB or the ice-shelf basal melting.

For assessing the computing times (are these wall-clock times?) reported in Sect. 4, some information about the used computer system and compiler would be nice. The reported "5 h 30 m" on p. 17, l. 15, lack any context and should be compared with the values reported earlier for the case of a constant resolution for the ice margin, grounding line and calving front.

Abstract l. 18 (also end of Sect. 4) vs. p. 2, l.11: I find this quite contradictory. If we take the statement seriously that a resolution of < 1 km around the grounding line is needed, then the example of a simulation for all major ice sheets with a 4-km resolution is insufficient. What happens if one really goes down to the required < 1 km? Won't

the computing times then become prohibitive?

Minor issues:

P. 2, l. 2/3: This justification does not really work. It is no problem to melt down a significant part of the Greenland ice sheet within only 10ˆ3 years under a decent climate warming. Same for West Antarctica, triggered by the marine ice sheet instability. The mentioned processes do not always require 10ˆ5-10ˆ7 years to become relevant.

P. 2. l. 30: I don't think that Elmer/Ice has an adaptive grid. Not sure about ISSM or MALI either. Please check this in the cited references. BISICLES (https://commons.lbl.gov/display/bisicles/BISICLES) definitely has, it should be mentioned in this context. BISICLES also employs finite-volume techniques.

P. 3, l. 4: "constitutes an optimum" is a pretty strong statement that implies that nobody will ever be able to do it better. Perhaps toning it down a bit; something like "we seek a compromise between these two families"?

P. 3, l. 5: "UFEMISM" for "Utrecht Finite Volume Ice-Sheet Model"? The abbreviation does not seem to fit very well, and the "FEM" in it rather alludes to the finite-element method.

P. 3, l. 16/17 (and again p. 6, l. 14): It is misleading to say that "Flow velocities for grounded ice are calculated using the shallow ice approximation". This applies only to the part due to internal deformation, while the part due to basal sliding is SSA. This is explained only later. Should be reformulated.

P. 3, l. 23: A reference should be provided for the Arakawa C grid.

P. 6, l. 14/15 and Eqs. (1) and (2): $u$, $v$ and $D$ do not only depend on $z$, but also on $x$, $y$ and $t$.

P. 7, Eq. (4): This form of the strain heating holds only for the SIA. What about the general form for SIA/SSA hybrid dynamics and the SSA for floating ice?

P. 7, Eq. (5): This equation holds for cold ice only. How is temperate ice treated? Just by cutting off temperatures exceeding the pressure melting point (not energy-conserving), or something more sophisticated?

P. 7, l. 21: I find it hard to believe that the stability is independent of the horizontal resolution. There is still horizontal advection in the equation, which is discretized explicitly. The time step of 10 years may be sufficient for all tested cases, but there will be a limit as one goes to higher resolutions.

P. 8, Eq. (7) and p. 9, Eq. (10): I would suggest to give up this separate Gamma factor and simply integrate it in the main equations.

P. 11, caption of Fig. 5: What is meant by "plotting artefact"?

P. 12, Fig. 6: The 50-km resolution can be nicely seen near the ice margin, which is fine. But I'm wondering why it cannot be seen equally well in Figs. 3, 4, 5 and 7? Different interpolators perhaps? If so, I'd suggest to use the same interpolation for all plots and, if it is not simple linear interpolation, it should be mentioned.

P. 13, Fig. 9; p. 14, Fig. 10: Perhaps different colours for the 20-kyr and 40-kyr cases? This would make the figures easier to read.

P. 14, l. 11: "MPI" should be defined.

P. 18, l. 33: Which value of this relaxation parameter omega have you actually used for the Antarctica tests?

P. 25/26, Appendix B: Apparently, a notation is used where subscripts (indices) x and y denote partial derivatives. This should be stated clearly. Further, some references to "Eq. 1" etc. appear that should probably be "Eq. B1" etc.

P. 25, l. 11: I don't understand this inequation. How can one compare the derivative of a quantity with the quantity itself? The units are even different.

P. 28, l. 3: The "solution" of the system of equations is the unknown vector of updated

temperatures. delta should rather be called the "right-side vector" (or "vector of the right sides").

---

## Author Comment (AC1) · 18 Feb 2021

Rebuttal to the reviews by Josefin Ahlkrona and Ralf Greve

We thank both reviewers for their insightful comments on the manuscript and would hereby like to address the concerns they raised. Comments in italics, below our rebuttal. Page and line numbers refer to the revised manuscript. We begin by addressing a few issues that were raised by both Josefin Ahlkrona (JA) and Ralf Greve (RG), starting with the two most important issues: the lack of benchmarking for the SSA part of the model, and the question of the usefulness of a SIA/SSA model in general.

JA: *"The model is only validated for the SIA equations, not the SSA equations. In order for the community to trust the accuracy of the model, the full SIA/SSA model must also be evaluated in experiments in some way. As the authors state, this is more difficult due to the lack of an analytical solution. However, the numerical discretisation can be tested by running convergence experiments, using a fine mesh solution as a reference. Does the errors decrease as expected when the mesh is refined?"*

RG: *"The most severe omission is that no attempt is made to verify the SSA part of the model, and the performance of the full model with respect to grounding line dynamics. Granted, analytical solutions are lacking. However, model intercomparisons have been done within the several MISMIP initiatives, most recently MISMIP+ by Cornford et al. (2020, doi: 10.5194/tc-14-2283-2020). To enable the reader to appreciate the performance of the new model, it would really be crucial to carry out such types of experiments and demonstrate how the model behaves in terms of grounding line advance and retreat as a function of resolution."*

We agree that we were overly hasty in dismissing the added value of performing benchmark experiments for the SSA. We have used the extension time kindly granted to us by the editor to perform a set of simulations of the first MISMIP experiment, which investigates grounding line migration under forced advanced/retreat cycles. Our results show a grounding line position that is resolution-independent (at least across the range of resolutions we investigated, 64 – 16 km) and displays no hysteresis during advance/retreat cycles. We achieved this by including the semi-analytical solution to the grounding-line flux for a Coulomb-type sliding law (Tsai et al., 2015) as a boundary condition to our SSA solver, which is common solution in several other ice sheet models. We have added a substantial paragraph describing this experiment to the Benchmark Experiments section of the manuscript. A more detailed description of the flux condition has been added to the Appendix where the SSA solver is described. We believe that this shows that our SSA solution behaves similarly to other ice sheet models.

**P13L10 - P14L13: Added a paragraph describing the newly performed MISMIP experiment.**
**P33L4 - P33L23: Added a description of the flux condition to the Appendix describing the SSA solution.**

JA: *"The SIA-SSA model is known to be inaccurate at the grounding line. A fine grid at the grounding line thus means that the numerical error is low, while the model error is inevitably high. Does it make sense to resolve the grounding line despite that the model errors remain, and if so, how much? Include experiments that shows the balance between model and numerical errors, by using a find grid full Stokes model as a reference solution. These experiments would be of value to all models using SIA-SSA at the grounding line."*
JA: *"Page 9, Line 10: Does it make sense to reduce the error to an error of 10 km, when the SIA is very inaccurate at the margin? How large is the model error?"*
JA: *"Page 18, Line 11-26: Try to use a fine resolution full Stokes MISMIP solution as a reference solution, in order to evaluate the SIA/SSA model and the convergence of the numerical scheme"*
RG: *"Abstract l. 18 (also end of Sect. 4) vs. p. 2, l.11: I find this quite contradictory. If we take the statement seriously that a resolution of < 1 km around the grounding line is needed, then the example of a simulation for all major ice sheets with a 4-km resolution is insufficient. What happens if one really goes down to the required < 1 km? Won't the computing times then become prohibitive?"*

These questions derive from three different but connected issues, which we feel should be answered together. The first is the validity of the SIA/SSA for certain parts of the ice sheet/shelf, the second is the ability of numerical models to correctly solve the SSA, and the third is the impact of resolution on non-numerical errors.

Regarding the first issue: it is true that the SIA and SSA, both being vertically

integrated approximations to the stress balance, are not valid in regions where the ice thickness is no longer negligible small compared to the horizontal features of the ice sheet and the underlying topography. This was shown quite nicely by the ISMIP-HOM experiments (Pattyn et al., 2008), although they only looked at instantaneous velocities and not at temporal ice-sheet evolution.

Regarding the second issue: the first MISMIP experiment showed that "standard" numerical methods for solving the SSA produced unsatisfactory results, which were strongly resolution-dependent (much more than would be expected based on numerical errors alone) and displayed significant grounding-line hysteresis (which is prohibited, as Weertman and Schoof have both proven that the SSA predicts a unique stable ice profile) unless model resolution was smaller than 100 m (an unachievable value for most practical purposes). However, it also showed that these problems could be greatly reduced by implementing a semi-analytical solution to the grounding-line flux as some sort of correction or boundary condition (with details varying across different models). This way, many models could achieve "good" results (i.e. weak resolution dependence and little hysteresis, though usually with a grounding-line position that still deviated significantly from the analytical solution) at much coarser resolutions (though exactly how coarse is difficult to derive from the literature, with different papers quoting values ranging from 1 km to 40 km). However, the validity of these kinds of "heuristic" approaches has been questioned, particularly because the semi-analytical solution is only valid in the absence of buttressing. We don't claim our model is doing better (or worse) than any of this class of models.

The third issue is that of non-numerical resolution errors, meaning the ability of a discrete grid to resolve spatial variability in a variable - in this case bedrock topography. Cuzzone et al. (2019, The Cryosphere) demonstrated the importance of this type of resolution error for Greenland. Many Greenland outlet glaciers will not be in contact with the ocean at all, if the resolution is too coarse to resolve the fjords they

lie in. Similarly, the present-day Antarctic grounding line has many areas of complex topography, with fjords funnelling ice flow into outlet glaciers, pinning points hampering the flow of ice shelves, etc. The very coarse (10-40 km) resolution typical of most palaeo-ice-sheet models cannot resolve such features. A small "hill" that could stabilise a grounding line won't be visible, leading to an overestimated ice-sheet retreat. The depth of an outlet fjord will be underestimated, leading to slower ice streams and a decreased sensitivity to buttressing losses. These errors will remain regardless of what kind of approximation to the stress balance is used, and whether or not that approximation is solved correctly by the numerical model. The dynamic adaptive grid we developed offers a clear advantage for these non-numerical resolution errors.

The question of whether or not an SIA/SSA-model is appropriate at all is one we deem to be beyond the scope of this study; a lot of work has been done on this already with the ISMIP-HOM experiments, and while we certainly agree that this is a very interesting topic, we do not want to join that discussion ourselves. Currently, higher-order or full-Stokes models are still too computationally expensive for palaeo-applications; right know, as far as we know, the "record" has been set by Cuzzone et al. 2019 (The Cryosphere) who simulated a section of the Greenland ice sheet during the Holocene, so about $10^4$ yr. This means that for now the SIA/SSA approach is the only feasible option for the kind of palaeo-ice-sheet modelling we're interested in. It might well be that some future researcher will develop a version of the stress balance that is valid for the entire ice sheet and yet still easy enough to solve to allow for long simulations, but until this happens, we will have to work with the approximations we have. With the new experiments we performed, our model performs similarly to other SIA/SSA models, in that it displays the desired qualitative behaviour (weak resolution dependence, no hysteresis; Pattyn et al., 2012, 2013, 2017), but does not provide a quantitatively correct solution. We are confident that the question of the validity of the heuristics that were used to achieve this performance, and how to adapt it to account for buttressing, will be answered some day, but again, for now we will assume that this

approach is good enough (like all other SIA/SSA-models implicitly assume). Where we believe our model to be an improvement over earlier models is the increased ability to resolve small topographical features that affect grounding-line dynamics. Even if, at some point in the future, a better way to numerically solve the SSA is invented, or an altogether better version of the stress balance is derived, this feature of our model will still be of added value.

We will expand the Discussion section of the manuscript to reflect this line of reasoning.
**P19L4 – P19L17: Added a paragraph to the Discussion about the SIA/SSA approach to grounding line dynamics.**
**P2L18 – P2L20: Stated the importance of resolving small-scale topographical features.**

JA: *"The Finite Volume Method is an unusual choice for an ice sheet model, and a motivation of this choice should be given. Add a summary of the method in the main paper, including a statement of the flux approximation (the specific details can remain in the appendix) and the order of the scheme. Discuss if numerical mass conservation is important compared to the model errors, preferably illustrated with experiments. As already mentioned in point 2), include convergence experiments. The convergence experiments in Figure 3-10 only show THAT the model converges - not if it converges with at the expected rate"*
JA: *"Page 7, Line 2-4: It could be potentially confusing to discuss Finite Difference schemes while the model in the paper is based on Finite Volumes. I suggest clarifying this paragraph by beginning the discussion with a couple of sentences explaining the conceptual difference between FD and FVM, then move on to describing how the flux is approximated, and lastly relate this to the FD references and Type I / Type II models"*
JA: *"Page 7, Line 2-4: I would like to see the formulas of ice fluxes and how they are used to calculated SIA velocities here, since the FVM is usually used for solving*

[Figure]

*equations rather than calculating formulas”*

JA: *“Page 7, Line 9: Add a summary of the FVM approach for solving the SSA”*

JA: *“Page 7: State the theoretical order of accuracy of the FVM scheme”*

JA: *“Section 2: Specify how the free surface variable h is updated”*

JA: *“Appendix A: Why did you choose to go through this derivation instead of employing some standard FVM scheme? Why did you choose to start from a finite difference scheme?”*

JA: *“Appendix A: As FVM is not commonly used in ice sheet modelling, write down the basics of finite volumes and describe your scheme in that framework.”*

RG: *“In Section 2.3, the short description of the ice-thickness solver only applies to the SIA, for which the ice-thickness equation can be written in diffusion form. How is the general ice-thickness equation (that includes the SSA part of the dynamics) solved?”*

JA: *“Page 3, Line 5: Why is the model abbreviated UFEMISM, instead of UFVMISM? The name suggests that it is a finite element model.”*

RG: *“P. 3, l. 5: "UFEMISM" for "Utrecht Finite Volume Ice-Sheet Model"? The abbreviation does not seem to fit very well, and the "FEM" in it rather alludes to the finite-element method.”*

The “finite volumes” in the name UFEMISM refer to the finite volume method that is used to integrate the ice mass continuity equation through time (the abbreviation “UFEMISM” instead of “UFVMISM” was chosen solely for ease of pronunciation). A description of this method was indeed lacking from the manuscript, and has been added as an Appendix to the revised version. This method uses the vertically averaged velocities on the grid cell boundaries to calculate the mass of ice leaving/entering a grid cell during a time step. The vertically averaged velocities are, following the approach of Winkelmann et al. (2011) in PISM, simply the sum of the velocities following from the SIA and the SSA (a weighted sum as in Bueler and Brown 2009 is also possible, but does not make much difference). The SIA velocities follow directly from local ice geometry according to Eqs. 1-2, all defined on the staggered mesh (i.e. the cell

boundaries) described in Appendix A. The way the SSA velocities are calculated (by using an iterative method to solve the differential equations) is described in another Appendix. How to name this method of solving the SSA is not entirely straightforward; both something like "finite differences on an unstructured grid" or "finite elements with explicitly derived linear basis functions" would fit our approach (since there is no fundamental difference between finite elements and finite differences, the choice of name, in practice, depends mostly on the modeller's choice of grid). It is definitely not solved with finite volumes, as this is a method for determining the temporal evolution of a conserved quantity given a certain flux distribution, rather than the instantaneous state of a variable.

We will clarify this in the Model Description – Overview section of the manuscript.

**P29L12 – P30L20: Added an Appendix detailing the finite volume approach.**
**P3L27: Added a justification for the name UFEMISM.**

For the revised manuscript, we have performed a small experiment to demonstrate the convergence of our discretisation scheme, showing that the discretisation errors in the first- and second-order partial derivatives decrease with mesh resolution to the second and first power, respectively. This experiment has been included at the end of Appendix A in the revised manuscript. We compare these results to those of a slightly different scheme based on least squares rather than average gradients (Syrakos et al., 2017), and find very similar discretisation errors for all resolutions. Syrakos et al. provide a theoretical derivation of the convergence of their scheme, which fits these results (i.e. 2nd-order for the first-order derivatives, and 1st-order for the second-order derivatives). We do not provide a similar derivation for our discretisation, but trust that the experimental results suffice. We have also expanded the derivation of the discretisation in Appendix A to include the complete expressions for the neighbour functions, to allow for easier comparison between the paper and the model source code. We have also added a figure demonstrating the convergence of the ice margin position with resolution in the EISMINT experiments. Here we see

first-order convergence, which is to be expected as the SIA is a diffusive equation, meaning that the change in ice thickness is dictated strongly by the curvature of the surface (a second-order partial derivative, which the earlier convergence experiment shows to be first-order convergent).

**P21L4 – P28L3: Included a more comprehensive derivation of our discretisation scheme in the Appendix.**

**P28L4 – P29L10: Added a description of two small convergence experiments to the Appendix.**

JA: *"Page 7: Add reference to strain heating form (equation 4)"*

RG: *"P. 7, Eq. (4): This form of the strain heating holds only for the SIA. What about the general form for SIA/SSA hybrid dynamics and the SSA for floating ice?"*

RG: *"P. 7, Eq. (5): This equation holds for cold ice only. How is temperate ice treated? Just by cutting off temperatures exceeding the pressure melting point (not energy-conserving), or something more sophisticated?"*

Our current thermodynamics scheme does not include any strain heating for shelves, and temperatures are indeed limited to the pressure melting point. We agree with the reviewers that this scheme is outdated. The reason we chose to implement it this way, is that we initially aimed to reproduce the physics of our current square-grid model IMAU-ICE, which uses the same scheme. Since the change from a square grid to an unstructured mesh presented a substantial overhaul of the model code, we aimed to keep the other parts functionally the same, to make it easier for ourselves to check that everything works as intended. Now that that's been established, we intend to update the thermodynamics (as well as several other model components, such as the surface climate and mass balance parameterisation, sub-shelf melt parameterisation, and basal hydrology and sliding) in the coming one or two years, and present these updates in another publication. We will clarify this in the Discussion section of the manuscript. Mind that for instance a model like Elmer is not solving thermodynamics

at all, so we feel it is justified to take our basic thermodynamic approach as a starting point.

**P20L4 – P20L6: Mentioned our intention to improve the thermodynamics scheme in the Discussion.**

JA: *"Include experiments showing how the mesh adaptivity works. The results shown in Figure 3-10 are done on a uniform grid. These should be done using mesh adaptivity, given that the mesh adaptivity is one of the main contributions of the paper. Experiment with choice of conditions for refinement in these experiments."*

JA: *"Page 4. Line 7-23: Do include a discussion the impact of different choices of conditions for refining the mesh, how do they impact the accuracy of different variables? These kind of conditions makes the mesh refinement interesting."*

JA: *"Page 5, Line 16: Is the partitioning updated as the mesh is updated to maintain proper load balancing? If so, how? If not, motivate why this is not needed."*

JA: *"Section 3: Elaborate on the purpose of each experiments Experiments section 3: These should be done using adaptive meshes."*

JA: *"Page 17, Line 11-26: Do this experiment with some of the conditions for refinement that were discussed on page 4 rather than explicitly setting the refinement"*

All of the experiments that are presented in the manuscript have used dynamic adaptive meshes, which are updated throughout the simulation based on modelled ice sheet geometry (which is where our approach differs from other ice-sheet models using fixed adaptive grids). We apologise if this was not clear from the text, and we will clarify this in the revised manuscript. We have added an Appendix to the revised manuscript that describes the mesh generation scheme in more detail, including a small experiment that demonstrates how the different prescribed resolutions are implemented, as well as a more detailed description of the parallelisation schemes used for both the mesh generation scheme, and the rest of the model.

**P35L8 – P38L17: Added an Appendix about mesh generation.**

JA: *"Page 2, Line 12: Include a reference to https://tc.copernicus.org/articles/10/307/2016/"*

This is an interesting study and we agree that it should be referenced. We will add it to the revised manuscript.
**P19L17: Mentioned this study in the newly added paragraph discussing the SIA/SSA approach.**

JA: *"Page 2, Line 15: Add a sentence explaining why fine resolution is more important when buttressing is significant."*

What we meant to say here, is that the "heuristic" solutions to the SSA resolution dependence might not be valid when buttressing is significant. If this is indeed the case, and these heuristics cannot be applied, then (right now) the only way to get "good" results with an SSA solver is by using an (unachievably) high resolution. We will clarify this in the manuscript.
**P2L7 – P2L20: Rephrased the paragraph discussing the resolution requirements for accurate grounding-line dynamics.**

JA: *"Page 2, Line 20: Discuss how the time step relates to the mesh size in a non-uniform mesh."*
JA: *"Section 2: Specify which equation is limiting the time step - the free surface or the temperature equation?"*
JA: *"Page 17, Line 8: A dynamic time step is mentioned here. An ellaboration on the dynamic time step for SIA/SSA should be added in the model description"*

We have added the equations for the adaptive time step to the new Appendix that describes the finite volume scheme for integration the ice mas continuity equation

through time. The time step for the thermodynamics module is independent of that of the ice dynamics, and is kept constant. We will clarify this in the manuscript. **P30L15 – P30L20: Added the equations for the adaptive time steps to the newly added Appendix about the finite volume approach.**

JA: *"Page 2, Line 28: Define what properly capturing grounding line dynamics means to you, given that the model equations you use do not include all stress components."*

In this context, "proper" grounding line dynamics means a grounding line position with a weak resolution dependence and no hysteresis (both being smaller than the uncertainties resulting from the climate, mass balance, bed topography, and other model forcings), as is common practise in non-full Stokes models. We will clarify this in the manuscript.
**P19L4 – P19L17: Added a paragraph to the Discussion about the SIA/SSA approach to grounding line dynamics.**

JA: *"Page 3, Line 4: Optimal in which sense? I believe it is a compromise rather than an optimum."*
RG: *"P. 3, l. 4: "constitutes an optimum" is a pretty strong statement that implies that nobody will ever be able to do it better. Perhaps toning it down a bit; something like "we seek a compromise between these two families"?"*

We agree that "compromise" is a more fitting phrase. We will change this in the manuscript.
**P3L9: changed "optimum" to "compromise".**

JA: *"Page 3, Line 20-21: Is there a limit to how fine mesh the user can choose? If so, why?"*

RG: *"What happens if one really goes down to the required < 1 km? Won't the computing times then become prohibitive?"*

As far as we have been able to find, there is no hard limit to the mesh resolution; we have performed very short schematic experiments at resolutions < 1km without trouble. In practice, computation time will be the main limiting factor in the palaeo-ice-sheet applications for which our model is intended. We will clarify this in the manuscript. **P37L4 – P37L7: Explained this in the newly added Appendix about mesh generation.**

JA: *"Page 3, Line 24: Elaborate on why this is important. Why is mass conservation important when the overall mass balance will never be accurate due to model errors and mesh resolution limits?"*

We have not experimented with different schemes for the time integration of the mass continuity equation. However, we have devoted a lot of work to the conservative remapping scheme that remaps the modelled ice thickness and englacial temperatures during mesh updates. We found that other schemes (nearest-neighbour, linear, quadratic) led to unacceptable amounts of numerical diffusion. In the Halfar and Bueler experiments this shows up quickly in the ice thickness profile, which tended to "flatten out"; in the EISMINT experiment (where the geometry is dictated by a fixed mass balance), this could be seen in the englacial temperature, which showed a persistent warm bias. The conservative remapping scheme we implemented is the only scheme we found that does not produce these errors. We do not know if similar phenomena will occur when a non-conservative solution to the mass continuity equation is used. We will clarify this in the manuscript.
**P38L10 – P38L17: Motivated the necessity for conservative remapping in the newly added Appendix about mesh generation.**

JA: *"Page 3, Line 26: Is the number of vertical layers hard coded to 15?"*
JA: *"Page 7, Line 20: Is 15 layers hard coded?"*
JA: *"Page 7, section 2.4: What is the spatial discretisation of the temperature equation?"*

The number and distribution of vertical layers is fully configurable; choosing 20, 50 or indeed any number of layers is up to the user. Increasing this number will of course slow down the thermodynamics module, particularly because at some point the fixed 10 yr time step will have to be reduced to maintain stability. The 15 layers used here were chosen during preliminary experiments, allowing for easy comparison with results from our other ice models (ANICE and IMAU-ICE), which also use 15 layers. The horizontal derivatives are discretised using the neighbour functions derived in Appendix A. We will clarify this in the manuscript.
**P3L32: Stated that the number of vertical layers is configurable.**

JA: *"Page 3, Line 30: There are other models that solve simplified equations suited for paleo models and also use mesh adaptivity, for instance the SIA/SSA mode of Elmer/Ice. Include references to these models."*

We discussed this with the Grenoble group very recently. From what we've been told, the SIA/SSA mode of Elmer/Ice still does not include a thermodynamical module, nor does it have a temporally adaptive mesh; meshes are generated outside the ice model, so while they can be tailored to the initial ice-sheet geometry, they are not updated during a simulation. Currently the only way to achieve this is to manually stop the simulation, project the data onto a new mesh, and restart. This is not suitable for palaeo-simulations. Only BISICLES includes a dynamic mesh, but it does not have the SIA/SSA ice dynamics and is not feasible for long palaeosimulations.

JA: *"Page 4, Line 12: Specify what is meant by "model content""*

Model content refers to what is encountered inside a certain triangle at a certain point in the simulation: land, grounded ice, floating ice, a grounding line, an ice margin, etc. This is now explained in more detail in the newly added Appendix about mesh generation.

JA: *"Page 5, Figure 1: The mesh extends into the ocean. Are there passive volumes outside the ice domain? Clarify."*

The mesh covers the entire model domain, which is chosen such that it envelops the ice sheet/shelf even during glacial conditions, without the ice margin reaching the domain boundary. Since the model must be able to simulate the inception of an ice sheet, the mass balance must be calculated over the entire domain, regardless of the presence of ice. In terms of ice dynamics, ice-free vertices can be considered "passive" in that they are skipped during calculations to improve computation speed. However, things like water depth, surface climate and mass balance are still calculated. We will clarify this in the newly added Appendix about mesh generation.

JA: *"Page 6, Line 10. Did you consider simply deforming the mesh instead of generating a new mesh, or using deformation to decrease the frequency with which the mesh has to be updated."*

We did not consider this. The current mesh generation algorithm requires an insignificant amount of computation time compared to the rest of the model, even during simulations with rapid changes in ice-sheet geometry (requiring more frequent mesh updates). Improving upon this does not have a high priority right now.

JA: *"Page 7, Line 1: The SIA is formulas to evaluate rather than equations to solve, since the velocity and pressure is already solved for."*

We will change this phrase in the manuscript.
**P5L4 – P5L8: Changed the phrasing.**

JA: *"Page 7: Write down not only the SIA formulas but also the SSA equations for consistency, together with a brief explanation of how they are merged."*

We will do so.
**P5L16 – P6L4: Added the SSA equations to the model description section.**

JA: *"Page 9, Line 6-7: This sentence needs clarification. As you are looking at numerical error, clarify how the surface slope impacts it so that it is clearer to the reader what is model error and numerical error"*

We agree that this sentence was poorly phrased. While the SIA itself might also not be valid at an ice margin (especially when the bed topography is not flat), the problem we refer to is the ability of numerical solutions of the SIA to capture the discontinuous surface slope predicted by the SIA continuum model (so indeed the numerical error rather than the model error). We will clarify this in the manuscript.
**P8L8: Changed this phrasing.**

JA: *"Figure 5-8: Can you include results from the EISMINT benchmark for in the figure for comparison?"*

Unfortunately, no. The original EISMINT experiments were done in 1996, and no original digital version of the publication or the data is available.

JA: *"Section 3.2: Explain the EISMINT experiment in such a way that a reader does not have to be familiar with the original publication. Describe e.g. how a fixed*

*versus moving margin is achieved"*

In the "fixed margin" experiments, the mass balance is such that the expected theoretical ice margin lies outside the model domain, and ice thickness at the boundary is artificially kept at zero. A moving margin is achieved by setting a zero mass balance integral over a bounded region fully enclosed within the model domain. We will clarify this in the manuscript.
**P9L12: Added a line describing the way a moving margin is achieved.**

JA: *"Results in Figure 9 and 10: Would it be possible to instead look at the margin, since this is where the resolution matters?"*

Such results are not available from the original EISMINT publication, so no intercomparison is possible here. Since these experiments describe a perfectly flat bed topography, we do not believe a finer resolution to be of added value here in any case.

JA: *"Page 15, Figure 12: Why is SIA so expensive?"*
JA: *"Page 15, Figure 12: In which part is the free surface update included?"*

The SIA is more expensive than the thermodynamics because it requires a much smaller time step to maintain numerical stability, especially at high resolutions. The computation time of the free surface update is included with the SIA. Since the surface is updated in every update of every model component (SIA, SSA, thermodynamics), and the SIA has by the smallest time step (especially at high resolutions), this seemed the obvious choice. We will clarify this in the manuscript.
**P16L17: Stated this.**

JA: *"Page 16, paragraph 1: Discuss why you think mesh update scales less well"*

All of the code written for mesh generation and updating, including the parallelisation scheme, was created from scratch by the authors. This has the advantage of resulting in code that is tailor-made for the ice-sheet model, using the same data and code structure and making it very easy to adapt the code to our needs and add new features (something that is not easy to do with commercial software packages). The drawback is that its performance is not optimal, particularly the parallelisation part. However, we are happy to report that we have very recently been granted funding to hire a software developer specialising in high-performance scientific computing, so we are confident that these issues will be resolved in the near future.
**P16L12: Stated this.**

JA: *"Page 16, Line 7: Move the definition of b_r to the next paragraph"*

Changed this.

JA: *"Page 17, Line 11-26: These results need to be presented more in detail in a figure or table, since this experiment leads to the result that UFEMISM is 10-30 times faster than ANICE"*

We have added a small table summarising the observed and estimated computation times for the different simulations with the two models.
**P18L10: Added a table listing computation times for the two models.**

JA: *"Page 3, Line 20: "Paradox" -> compromise."*

Rephrased this.

JA: *"Page 3, Line 21: "config file" -> "configuration file""*

Changed this.

JA: *"Page 4, Line 2: "refinement " -> refinement"*

Changed this.

JA: *"Page 4, Line 17-20: The sentence starting with "Other conditions" is a bit unclear and long"*

Changed this.

JA: *"Page 10, Equation 10: Add spaces between formulas"*

Changed this.

JA: *"Page 11, Table 1: Order the rows so that they are consistent with the order of the figures, or vice versa"*

Unfortunately there is no nice way to do this without causing even greater confusion; the numbering of the experiments is fixed in the 1996 publication, and presenting them in a different order in our paper would not make sense.

JA: *"Page 12, Line 14: Vertical discretisation -> Vertical resolution"*

Changed this.

JA: *"Page 14, Line 11: "shared memory" -> shared memory"*

[Figure]

Changed this.

JA: *"Figure 12: Include in the caption an explanation of what bc and br are."*

Changed this.

RG: *"In Section 4, the description of the experiments, given in a mere four lines (p. 14, l. 15-18), lacks detail. What are the initial conditions for the experiments? What are the physical parameters (rate factor, basal sliding law, heat conductivity & capacity, geothermal heat flux, etc.)? What is assumed for ice-shelf basal melting? The tested resolutions should be mentioned in the text, not only in the figure captions. What happens to the ice sheet in a control scenario (no warming applied)? I am actually quite surprised that the ice sheet reacts so strongly on just a surface warming without (I presume) changes in the SMB or the ice-shelf basal melting."*

The experiment described in Section 4 is not intended to present a realistic depiction of possible future Antarctic retreat. It only serves to demonstrate computational performance of the model. The rapid retreat is indeed not caused by the temperature change itself, but by the strong increase in surface melt resulting from the insolation-temperature based mass balance parameterisation we included. We will clarify this in the manuscript.
**P15L5 – P15L9: Did this.**

RG: *"For assessing the computing times (are these wall-clock times?) reported in Sect. 4, some information about the used computer system and compiler would be nice. The reported "5 h 30 m" on p. 17, l. 15, lack any context and should be compared with the values reported earlier for the case of a constant resolution for the ice margin, grounding line and calving front."*

The simulations described in Sect. 4 were run on the LISA computer cluster operated by SURFsara, using an Intel Xeon Gold 6130 Processor with a 2.1 GHz clock frequency and 22 MB cache, and were compiled with the iFort compiler. We will clarify this in the manuscript.
**P15L15: Did this.**

The number of 5h 30m is specific for the simulation described in the second half of Sect. 4, which was created to resemble a typical glacial cycle simulation as closely as possible (running a full glacial cycle is not yet feasible, as this requires some more work on the climate forcing) so that it can be meaningfully compared to the computation time of ANICE. Comparing this number with the earlier numbers for the simulations where the grounding line, ice margin, and calving front all had the same resolution, is not meaningful since those model settings are not likely to be used in a realistic simulation.

RG: *"P. 2, l. 2/3: This justification does not really work. It is no problem to melt down a significant part of the Greenland ice sheet within only 10Ëʟ3 years under a decent climate warming. Same for West Antarctica, triggered by the marine ice sheet instability. The mentioned processes do not always require 10Ëʟ5-10Ëʟ7 years to become relevant."*

We agree that not all interactions in the Earth system require such a long time to become relevant. We will change "many processes" to "several processes".

RG: *"P. 2. l. 30: I don't think that Elmer/Ice has an adaptive grid. Not sure about ISSM or MALI either. Please check this in the cited references. BISICLES (https://commons.lbl.gov/display/bisicles/BISICLES) definitely has, it should be mentioned in this context. BISICLES also employs finite-volume techniques."*

Elmer/Ice, ISSM and MALI have spatially adaptive grids, whereas UFEMISM and BISICLES have spatially and temporally adaptive grids. We will clarify this in the manuscript, and use the phrases "static adaptive grid" and "dynamic adaptive grid" to differentiate between the two approaches.
**P2L34 – P3L3: Defined the difference between static and dynamic adaptive grids.**

RG: *"P. 3, l. 16/17 (and again p. 6, l. 14): It is misleading to say that "Flow velocities for grounded ice are calculated using the shallow ice approximation". This applies only to the part due to internal deformation, while the part due to basal sliding is SSA. This is explained only later. Should be reformulated."*

Changed this.

RG: *"P. 3, l. 23: A reference should be provided for the Arakawa C grid."*

Did this.

RG: *"P.6,l. 14/15 and Eqs. (1) and (2): u,v and D do not only depend on z, but also on x,y and t."*
Clarified this.

RG: *"P. 7, l. 21: I find it hard to believe that the stability is independent of the horizontal resolution. There is still horizontal advection in the equation, which is discretized explicitly. The time step of 10 years may be sufficient for all tested cases, but there will be a limit as one goes to higher resolutions."*

We agree that the phrasing "for all resolutions" is inaccurate. We will change this.
**P3L29: Added a reference to Arakawa and Lamb, 1977.**

RG: *"P. 8, Eq. (7) and p. 9, Eq. (10): I would suggest to give up this separate Gamma factor and simply integrate it in the main equations"*

The gamma factor is present in the original formulation of the exact solution by Bueler et al., 2005. We believe removing it would lead to more confusion.

RG: *"P. 11, caption of Fig. 5: What is meant by "plotting artefact"?"*
RG: *"P. 12, Fig. 6: The 50-km resolution can be nicely seen near the ice margin, which is fine. But I'm wondering why it cannot be seen equally well in Figs. 3, 4, 5 and 7? Different interpolators perhaps? If so, I'd suggest to use the same interpolation for all plots and, if it is not simple linear interpolation, it should be mentioned."*

This has indeed got to do with the interpolation from an unstructured grid to an equally-spaced line for plotting, which is not always easy to do, especially for a non-conserved, discontinuous quantity such as ice velocity. In the moving margin cases, some "smoothing" is applied by creating multiple transects along different radians (exploiting the circular symmetry of the ice sheet) and averaging them out. An earlier version of this scheme causes some "bumps", which we have since fixed. We will update the Figure.
**Updated Figs. 7 and 8.**

RG: *"P. 13, Fig. 9; p. 14, Fig. 10: Perhaps different colours for the 20-kyr and 40-kyr cases? This would make the figures easier to read."*

This is a good suggestion. We will do this.
**Updated Figs. 9 and 10.**

RG: *"P. 14, l. 11: "MPI" should be defined."*

Did this.

RG: *"P. 18, l. 33: Which value of this relaxation parameter omega have you actually used for the Antarctica tests?"*

These simulations were performed with a spatially uniform over-relaxation parameter of 1.1. We will state this in the Appendix detailing the SSA solution scheme.

RG: *"P. 25/26, Appendix B: Apparently, a notation is used where subscripts (indices) x and y denote partial derivatives. This should be stated clearly. Further, some references to "Eq. 1" etc. appear that should probably be "Eq. B1" etc."*

Fixed this.

RG: *"P. 25, l. 11: I don't understand this inequation. How can one compare the derivative of a quantity with the quantity itself? The units are even different."*

This is based on the approach by Determann (1991) and Huybrechts (1992), where the lateral variations of the effective strain rate deta/dx, deta/dy are neglected. Determann (1991) and Huybrechts (1992) show that, since these terms are small compared to variations of the individual strain rates, this does not significantly affect the solution, while improving numerical stability and computational efficiency. We will clarify this in the manuscript.
**P31L9 – P31L13: Explained this simplification.**

RG: *"P. 28, l. 3: The "solution" of the system of equations is the unknown vector of updated temperatures. delta should rather be called the "right-side vector" (or "vector of the right sides")."*

**GMDD**

Fixed this.

---

## Author Response (AR2)

**Rebuttal to the reviews by Josefin Ahlkrona and Ralf Greve**

We thank both reviewers for their insightful comments on the manuscript and would hereby like to address the concerns they raised. Comments in italics, below our rebuttal.

*JA: The mesh adaptivity is the main feature of the manuscript, and this should be advertised better in the manuscript. The authors added a text on how the mesh adaptivity is used in the benchmark experiments, but I think it is still easy to miss this point. A graphical representation of the mesh used in the benchmark experiments would help. The point of the mesh adaptivity should also be made even clearer in the introduction /abstract: i.e. that despite that it is impossible to resolve dynamics due to the model, there is still a point in resolving topography/data.*

We will explicitly state that the mesh is adapted to the modelled ice-sheet geometry during a simulation in the abstract, and we will add a line stating the main advantage of a locally high resolution as being able to better resolve topographical/other features. We will also add a line to the first paragraph of Sect. 3 (Model verification and benchmark experiments) stating that all benchmark experiments were performed with dynamic adaptive meshes. Lastly, we will briefly mention the dynamic mesh adaptivity in Sect. 2.2 (Unstructured triangular mesh), referring to Appendix E for details (specifically to Fig. E3, which shows a set of different meshes for the Antarctic retreat simulation and thereby illustrates to the reader what a dynamic adaptive mesh means in practice).

*JA: "Motivate why you benchmark the SIA and SSA separately, instead of benchmarking your complete hy- brid model."*

The newly added MISMIP experiment uses the complete hybrid SIA/SSA approach, not just the SSA. We will clarify this in the manuscript.

*JA: "Discuss what the need for the semi analytical solution to the grounding line flux means for the accu- racy of practical applications."*

The most important drawback of the GL flux condition is likely still the poor representation of buttressing. No studies have yet investigated just how large an effect this particular model error has on palaeo-ice-sheet dynamics with respect to the uncertainties arising from proxy data, paleoclimate forcing, and other physical processes. We will add a line to the Discussion section to reflect this.

*JA: "Line 21: "pixels" -¿ grid points?"*

We will correct this.

*JA: "Equation B6a-b: I would prefer these equations in the main manuscript. Mention what Dc is in a place close to this equation."*

We will move these equations to Sect. 2.3 (Model description - Ice dynamics).

> *RG: "On the abbreviation "UFEMISM": I still don't like it. Every knowledgeable reader will immediately assume that it is a finite-element model, and this is unnecessarily misleading. Granted, "UFVMISM" is not so nice to pronounce, but still, this is not good... Should be reconsidered."*

The discussion about the (lack of a) fundamental distinction between finite elements and finite differences has been going on for a long time. Deriving a system of linear equations representing a PDE using finite elements (with linear basis functions) on a regular square grid yields the same result as using finite differences, which suggests that FD is simply a special case of FEM (although many people claim that there still is a fun- damental difference in interpretation, and the identical equations are merely coincidental). In practise, it seems that the choice of name correlates mostly with the choice of grid; discretising and solving a PDE on a regular square grid is called finite differencing, whereas doing it on an unstructured mesh is called finite elements. In our experience, most people associate the phrase "finite elements" with irregular triangles/polygons, and "finite differences" with regular squares. We therefore believe our model name to be appropriate.

> *RG: "P. 5, Eqs. (1), (2), and explaining text: The quantity D shouldn't be called a "diffusivity" as it does not appear in any diffusion equation. What is usually called diffusivity D in this context is the depth integral of your D (e.g., Huybrechts et al. 1996, Greve and Blatter 2009 ["Dynamics of Ice Sheets and Glaciers", Springer]). This is the quantity that appears as a diffusivity in the SIA version of the ice thickness equation. However, it is irrelevant in your context as you don't have pure SIA dynamics, so that you must solve the general form of the ice thickness equation (your Eq. (B1))."*

We will change the phrasing of the text accordingly.

> *RG: "P. 5, Eqs. (1): The variable zeta should be defined (appears only later in Eq. (D1)). As the non-transformed integral goes from b to z [e.g., Eq. (5.93) by Greve and Blatter (2009) without the contribution from Weertman sliding that you don't have], the transformed integral must go from zeta to 1, and the integration variable should be called zeta' or zeta-bar rather than zeta."*

We agree that the letter $\zeta$ was a confusing choice for the dummy integrand; we will change this to $z'$. Also, the integral should indeed have been from $z' = b$ to $z' = h$; we will fix this.

> *RG: "P. 6, l. 5-8: Related to the above said, classifying your model as a Type I model in the sense of Huybrechts et al. (1996) is not appropriate. It does not fit this classification pattern at all as you do not solve the diffusive SIA version of the ice thickness equation."*

We agree that the reference to Huybrecht's et al.'s SIA-only model classification is confusing. We will re- move it.

*RG: "P. 6, Eqs. (6),(7), and explaining text: As for the strain heating, it is not sufficient to say that this is "for grounded ice only". The form you give in Eq. (7) is only valid for the SIA. Since you have hybrid SIA/SSA dynamics for grounded ice rather than SIA, it is an additional simplification to assume that the strain heating is SIA-type. This should be stated clearly."*

We will clarify this in the manuscript.

*RG: "P. 15, l. 3-9: I understand that the experiment, in the authors' words, "is not intended to present a realistic depiction of possible future Antarctic retreat. It only serves to demonstrate computational performance of the model." Nevertheless, I think it should be described in more detail to allow the readers assessing what is going on. Quoting my original review: "What are the initial conditions for the experiments? What are the physical parameters (rate factor, basal sliding law, heat conductivity and capacity, geothermal heat flux, etc.)? What is assumed for ice-shelf basal melting?" If the authors don't want to have this in the main text, it may be put in an appendix section."*

We will add this information to the manuscript.